**SOFTWARE**

# gapseq: informed prediction of bacterial metabolic pathways and reconstruction of accurate metabolic models

Johannes Zimmermann[1], Christoph Kaleta[1] and Silvio Waschina[1,2]*

*Correspondence:
s.waschina@nutrinf.uni-kiel.de
[1]Christian-Albrechts-University Kiel,
Institute of Experimental Medicine,
Research Group Medical Systems
Biology, Michaelis-Str. 5, 24105 Kiel,
Germany
[2]Christian-Albrechts-University Kiel,
Institute of Human Nutrition and
Food Science, Nutriinformatics,
Heinrich-Hecht-Platz 10, 24118 Kiel,
Germany

## Abstract

Genome-scale metabolic models of microorganisms are powerful frameworks to predict phenotypes from an organism's genotype. While manual reconstructions are laborious, automated reconstructions often fail to recapitulate known metabolic processes. Here we present gapseq (https://github.com/jotech/gapseq), a new tool to predict metabolic pathways and automatically reconstruct microbial metabolic models using a curated reaction database and a novel gap-filling algorithm. On the basis of scientific literature and experimental data for 14,931 bacterial phenotypes, we demonstrate that gapseq outperforms state-of-the-art tools in predicting enzyme activity, carbon source utilisation, fermentation products, and metabolic interactions within microbial communities.

**Keywords:** Metabolic pathway analysis, Metabolic networks, Genome-scale metabolic models, Benchmark, Community simulation, Microbiome, Metagenome

## Background

> Anything you have to do repeatedly
> may be ripe for automation.
>
> — Doug McIlroy

Metabolism is central for organismal life. It provides metabolites and energy for all cellular processes. A majority of metabolic reactions are catalysed by enzymes, which are encoded in the genome of the respective organism. Those catalysed reactions form a complex metabolic network of numerous biochemical transformations, which the organism is presumably able to perform [1].

In systems biology, the reconstruction of metabolic networks plays an essential role, as the network represents an organism's capabilities to interact with its biotic and abiotic environment and to transform nutrients into biomass. Mathematical analysis has shown great potential for dissecting the functioning of metabolic networks on the level

of topological, stoichiometric, and kinetic models [2], which together provide a wide array of methods [3]. Although different microbial metabolic modelling approaches exist, they can be summarised by a theoretical framework that provides a unifying view on microbial growth [4]. Metabolic models not only have demonstrated their ability to predict phenotypes on the level of cellular growth and gene knockouts, but also provide potential molecular mechanisms in form of gene and reaction activities, which can be validated experimentally [5–7]. Due to this predictive potential, metabolic models have been applied to identify metabolic interactions between different organisms [8–13], to study host-microbiome interactions [14–16], to predict novel drug targets to fight microbial pathogens [17, 18], and for the rational design of microbial genotypes and growth-media conditions for the industrial production or degradation of biochemicals [19, 20]. Furthermore, recent advances in DNA-sequencing technologies have led to a vast increase in available genomic- and metagenomic sequences in databases [21], which further expands the applicability of genome-scale metabolic network reconstructions.

In the process of genome-scale metabolic network reconstruction, the genomic content of an organism is linked to biochemical processes, including enzymatic reactions and cross-membrane metabolite transport [22]. Therefore, the quality and integrity of network models depend on the genome sequence annotation and the underlying reaction and transporter database [22, 23]. Advances in the computational annotation of genomes and the massive increase of biochemical knowledge stored in online databases [24–26] have prompted the development of several software approaches to automate the reconstruction process [27]. A recent study by Mendoza et al. comprehensively compared seven current genome-scale metabolic reconstruction tools [28], namely AuReMe [29], CarveMe [30], Merlin [31], MetaDraft [32], ModelSEED [33], Pathway Tools [34], and RAVEN [35]. On the basis of 18 specific criteria, Mendoza et al. concluded that each tool displayed strengths and shortcomings in different aspects [28]. One of the comparison criterion was the ability of the software to provide a 'ready-to-use' model as output, where the 'use' refers to the possibility to perform flux balance analysis (FBA [36]) or FBA-derived simulation techniques to predict the organism's metabolic physiology, including biomass production, under a given chemical environment. This criterion was fully met only by CarveMe and ModelSEED [28].

The feature to directly obtain network models that can be used for FBA-based growth simulations is especially powerful in situations where large numbers of new microbial genomes are assembled from high-throughput metagenomic datasets [37]. In such studies, the models can be used to predict physiological properties of the sampled microbial community, including metabolite cross-feeding interactions between species. However, a fundamental issue with automatically reconstructed genome-scale models is that their physiological predictions (e.g. using FBA) are often inaccurate [38]. Since the reconstruction process involves various steps, the causes for false metabolic flux predictions from automatic reconstructions can be manifold: First, inconsistencies in databases can lead to an incorporation of imbalanced reactions into the metabolic network, which may become responsible for incorrect energy production by futile cycles [22]. Second, many genes are lacking a functional annotation due to a lack of knowledge [39] and, thus, also the gene products cannot be integrated into the metabolic networks, which potentially lead to gaps in pathways. And third, the gap-filling of metabolic networks is frequently done by adding a minimum number of reactions from a reference database that facilitate growth under a

chemically defined growth medium [33, 40, 41]. Such approaches miss further evidences potentially hidden in sequences and are biased towards the growth medium used for gap-filling.

The potential of automated reconstruction tools to directly predict metabolic-physiological properties of organisms based on their genome sequence was so far only evaluated on the basis of smaller experimental data sets from model laboratory strains such as *Escherichia coli* K12 or *Bacillus subtilis* 168. The overall performance of reconstruction tools, particularly for non-model organisms, is therefore insufficiently assured. Yet, accurate phenotype predictions for a wide range of organisms is crucial for the broad application of automated network reconstruction pipelines in research. For instance, genome-scale metabolic network reconstructions are increasingly applied to simulate complex metabolic processes in microbial communities [42, 43]. Such simulations are highly sensitive to the quality of the individual metabolic networks of the community members. This is because the accurate prediction of by-products and carbon source utilisation is crucial for the correct prediction of metabolic interactions since the substances produced by one organism may serve as resource for others [44]. Thus, in multi-species communities, the metabolic fluxes of organisms are intrinsically connected, which can lead to error propagation when one defective model affects otherwise correctly working models. As a consequence, the feasibility of community modelling fundamentally depends on the accuracy of the individual organismal models.

In this work, we present gapseq a novel software for pathway analysis and metabolic network reconstruction. The pathway prediction is based on multiple biochemistry databases that comprise information on pathway structures, the pathways' key enzymes, and reaction stoichiometries. Moreover, gapseq constructs genome-scale metabolic models that enable FBA-based metabolic phenotype predictions as well as the application in simulations of community metabolism. Models are constructed using a manually curated reaction database that is free of energy-generating thermodynamically infeasible reaction cycles. As input, gapseq takes the organism's genome sequence in FASTA format, without the need for an additional annotation file. Network topology as well as sequence homology to reference proteins inform the filling of network gaps. A novel Linear Programming (LP)-based gap-filling algorithm identifies and resolves gaps in order to enable biomass formation on a given medium. In addition, the algorithm also identifies and fills gaps in metabolic functions, whose presence in the network is supported by sequence homology to reference proteins and which are likely to be relevant for growth in environments that are different to the chosen gap-filling medium. This approach reduces the gap-filling medium-specific effects on the final network structures and thereby increases the versatility of gapseq models for subsequent physiological predictions under various chemical growth environments. Finally, we use large-scale phenotype data sets to validate enzyme activity, carbon source utilisation, fermentation products, gene essentiality, and metabolite cross-feeding interactions in microbial communities. The results obtained with gapseq are benchmarked against CarveMe [30] and ModelSEED [33], as these tools also provide the full procedure to construct models, which can directly be employed for FBA-based metabolic flux simulations of microbial growth.

## Results

### Biochemistry database and universal model

The pathway, transporter, and complex prediction is based on a protein sequence database that is derived from UniProt as well as TCDB and consists in total of 131,207 unique sequences (112,056 reviewed unipac 0.9 clusters and 19,151 TCDB transporter) and also 1,138,176 unreviewed unipac 0.5 cluster that can be included optionally. The reference protein sequences are regularly updated by the gapseq maintainers using the latest UniProt and TCDB releases. gapseq automatically checks for updates and retrieves the latest reference sequences upon start of the software. For the construction of genome-scale metabolic network models we have built a biochemistry database, that is derived from the ModelSEED biochemistry database. In total, the resulting curated gapseq metabolism database comprises 15,150 reactions (including transporters) and 8446 metabolites. All metabolites and reactions from the biochemistry database are incorporated in the universal model that gapseq utilises for the gap-filling algorithm. If all dead-end metabolites and corresponding reactions would be removed, the universal model comprises 10,792 reactions and 3885 metabolites. However, since genome-scale metabolic networks are also used as structured knowledge-bases, no dead ends are removed from the universal model. It needs to be noted, that the current biochemistry database and the derived universal model represents mainly bacterial metabolic functions and that, at the current version of gapseq, the database does not include all archaea-specific nor eukaryotic-specific reactions. However, those reactions and, thus, also the possibility to use gapseq for the reconstruction of archaeal and eukaryotic models will be included in a later version of the software.

### Enzymatic data

Microbial isolates are commonly subject to laboratory enzyme activity tests for strain characterisation and identification. The Bacterial Diversity Metadatabase (BacDive) provides results from enzyme activity tests spanning a wide taxonomic range and different enzymes [45]. This data represents highly valuable phenotypic information that can be used to scrutinise whether metabolic network models of microorganisms also harbour the enzymatic reaction that was experimentally tested. Here, we performed this evaluation for automated network reconstructions obtained with the tools CarveMe [30], ModelSEED [33], and our gapseq approach.

   In total, we compared 10,538 enzyme activities, which consists of data for 3017 organisms and 30 unique enzymes. For all organisms, genome-scale metabolic models were constructed using the three different software tools. gapseq models had with 6% the lowest false negative rate compared to CarveMe (32%) and ModelSEED (28%). Correspondingly, gapseq showed with 53% also the highest true positive rate compared to CarveMe (27%) and ModelSEED (30%), while the rates of false positive and true negative predictions were comparable (Fig. 1a). For this test, the most prominent EC numbers were the catalase, 1.11.1.6, accounting for 26% of the comparisons and the cytochrome oxidase, 1.9.3.1, accounting for 22%, which reflects the ecological importance of cytochrome oxidases and catalases as proxy for an aerobic lifestyle. The overall results remain stable when sampling equal numbers of test data for each EC number and thereby controlling for a potential bias by the over-representation of these EC numbers (Additional file 1: Fig. S4).

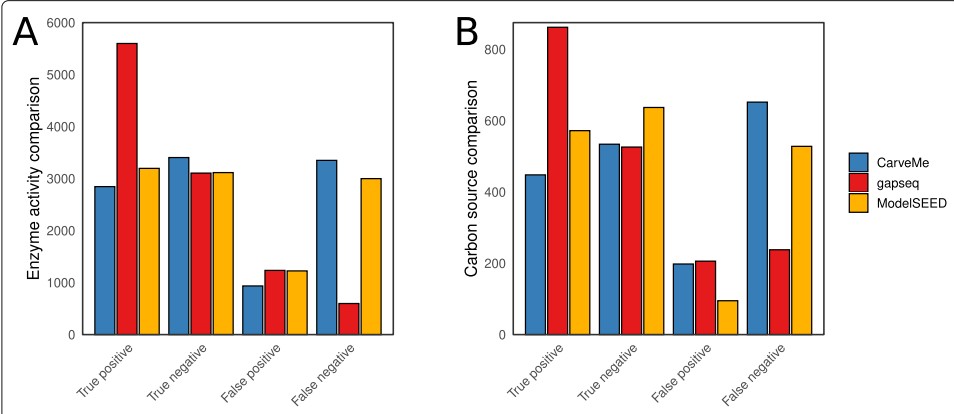

**Fig. 1** Results from enzyme activity and carbon source validations. **a** In total 10,538 enzyme activities (30 enzymes and 3017 organisms) of experimental data from the DSMZ BacDive database [45] were compared for three different methods. **b** The predictions of 1795 carbon sources (48 unique carbon sources and 526 organisms) were evaluated with data from the ProTraits database [46]

## Carbon source utilisation

The bacterial kingdom comprises a tremendous diversity in carbon source utilisation strategies. In the context of genome-scale metabolic modelling, a major challenge is the accurate prediction of carbon source utilisation phenotypes from an organism's genome sequence. In order to evaluate gapseq's potential to predict carbon source utilisation capabilities we retrieved data on bacterial phenotypes from the ProTraits resource [46]. In brief, ProTraits provides information on phenotypic traits, including carbon source utilisation, of individual microorganisms, where the phenotypic trait data is inferred from scientific literature and comparative genomics. Here, we evaluated the quality of automated model reconstruction pipelines by testing if the models are able to recapitulate carbon source utilisation phenotypes as indicated in ProTraits.

In summary, we compared 1795 different carbon source utilisation predictions for 526 organism and 48 carbon sources (Fig. 1b). gapseq outperformed the other methods in terms of false negatives (13% compared with 29% ModelSEED and 36% CarveMe) and true positives (47% compared with 31% ModelSEED and 24% CarveMe). ModelSEED showed fewer false positives (5% compared with 11% gapseq and 11% CarveMe) and more true negatives (35% compared with 29% gapseq and 29% CarveMe). gapseq, predicted most false positives for formate (29 times). This overestimate of formate as potential carbon source is likely due to the fact that we tested carbon source utilisation on the basis of electron transfer from the source to electron carriers (i.e. ubiquinol, menaquinol, or NADH), which is analogous to the experimental carbon source test of BIOLOG plates [47]. However, while it is known that formate can serve in fact as electron donor in a number of different bacteria [48], the role as source of carbon atoms for the synthesis of biomass components is limited to a few known methylotrophs [49]. Across all methods, the most accurately predicted carbon sources, with more than 100 tested organisms, were fructose (92% correct predictions), mannose (91%), or arginine (82%), whereby the predictions were less accurate for arabinose (29% correct predictions), dextrin (41%), or acetate (51%).

In general, we note that testing carbon source utilisation via the proxy of electron transfer from the substrate to reducing equivalents has the advantage that one can test a vast

number of model reconstructions without the need to define a complete chemical growth environment that contains besides the carbon source also all other compounds required for growth (e.g. specific amino acids in case of auxotrohies). However, this approach has the shortcoming that in some cases, the ability of an organism to use a substance as electron donor does not always imply that the substance can also be used as source of carbon. Nevertheless, we argue that the implemented carbon source utilisation prediction is pertinent as it reflects the same approach as BIOLOG plates, which is an established system for carbon source utilisation profiling.

### Gene essentiality

We compared the ability of `gapseq` models to predict the essentiality of genes with predictions from ModelSEED and CarveMe reconstructions as well as with curated models for the same organisms (Fig. 2). As expected, the curated models outperformed all three automated reconstruction tools for most species and prediction metrics (namely precision, sensitivity, specificity, accuracy, and F1-score). Interestingly, for *Shewanella oneidensis* and *Pseudomonas aeruginosa* `gapseq` reconstructions outperformed curated models in most test scores with the exceptions of the sensitivity in the case of *S. oneidensis* and specificity for *P. aeruginosa* (Fig. 2c, d). Compared to CarveMe, `gapseq` showed in four out of five cases a higher sensitivity in essentiality predictions but, at the same time, a slightly lower specificity. This pattern is attributed to the fact that `gapseq` models tend to predict more genes as essential than CarveMe, leading to a higher number of true positive (TP) predictions but also more false positives (FP). For most organisms and on the basis of most prediction metrics, `gapseq` outperformed network models that were

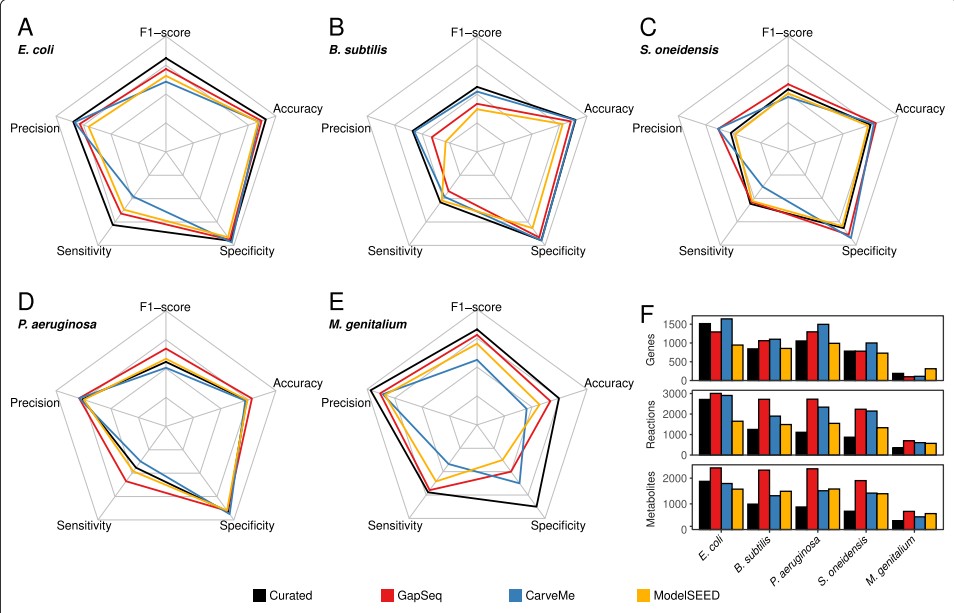

**Fig. 2** Results from model gene essentiality tests for five bacterial species. **a** *Escherichia coli*. **b** *Bacillus subtilis*. **c** *Shewanella oneidensis*. **d** *Pseudomonas aeruginosa*. **e** *Mycoplasma genitalium*. Results from `gapseq` models (red) are compared to CarveMe (blue) and ModelSEED (yellow) models, as well as to published curated genome-scale metabolic models (black) of the respective organisms. Radar chart axes scales are linear with 0 in the centre and 1 at the corners. **f** Counts of genes, reactions (including exchanges and transporters), and metabolites in each reconstruction

reconstructed using ModelSEED. The results presented here consider genes as essential, if the predicted growth rate of the focal gene-knockout strain was below 0.01 h$^{-1}$. However, we note that the results remained virtually unaltered with a higher (0.05 h$^{-1}$) or lower (0.001 h$^{-1}$) threshold (Additional file 1: Fig. S1).

Accurate gene essentiality predictions rely on precise gene-protein-reaction (GPR) associations, which are formulated as Boolean expressions to describe the reactions' dependence on proteins and the corresponding protein-encoding genes. The automated prediction of GPR associations is especially challenging for reactions that depend on protein complexes consisting of different protein/peptide subunits. We compared the GPR expressions for such reactions in the metabolic network of *E. coli* between the manually curated network (iML1515) and the automated reconstructions from CarveMe, ModelSEED, and gapseq (Additional file 2: Table S6). 59 protein complex-associated reactions were shared among all networks. Considering the GPR associations of the curated network as reference, only 6% were equivalent to those in the CarveMe network, 10% for ModelSEED, and 19% for gapseq. These results suggest, that accurate GPR association predictions are still a weakness in the tested automated reconstruction tools and thereby limit the essentiality predictions of individual genes, which encode protein subunits.

### Fermentation products

Anaerobic or facultative anaerobic bacteria utilise different fermentation pathways in order to extract energy from environmental compounds by chemical transformations in the absence of oxygen. We tested if fermentation products can be predicted by metabolic reconstructions obtained from gapseq, CarveMe, and ModelSEED for 24 different bacterial organisms (Fig. 3). The organisms were selected based on following criteria: (1) the organisms have a published RefSeq genome sequence, (2) are known anaerobic or facultative anaerobic organisms, and (3) the identity of fermentation products has been experimentally described and reported in primary literature (Additional File 2: Table S2). Overall, gapseq showed the highest number of true positive predictions (TP) with 50 TP predicted with the Minimise-Total-Flux (MTF) and 51 TP predicted with Flux-Variability-Analysis (FVA) which is substantially higher compared to CarveMe (15 TP with MTF, 16 TP with FVA) and ModelSEED (2 TP, 4 TP). The production of the short-chain fatty acids acetate, butyrate, and propionate was correctly predicted (TP) by gapseq in 91% of cases and thereby outcompetes CarveMe (12%) and ModelSEED (0%), which did not predict butyrate or propionate production for any tested organism. Moreover, gapseq correctly predicted homolactic fermentation by *Lactobacillus delbrueckii* and *Lactobacillus acidophilus*, which is dominated by lactate as fermentation end product and also predicted known heterolactic fermentation by *Bifidobacterium longum*, *Bifidobacterium animalis*, and *Lactobacillus plantarum*. However, gapseq failed to predict lactate production of organisms that utilise different fermentation strategies, which also yield lactate (e.g. mixed-acid fermentation by *Escherichia coli*). Interestingly, the predicted quantities of fermentation product release is higher for true positive than for false negative predictions (Fig. 3). This further suggests, that gapseq is able to predict the main fermentation products of bacterial organisms during anaerobic growth.

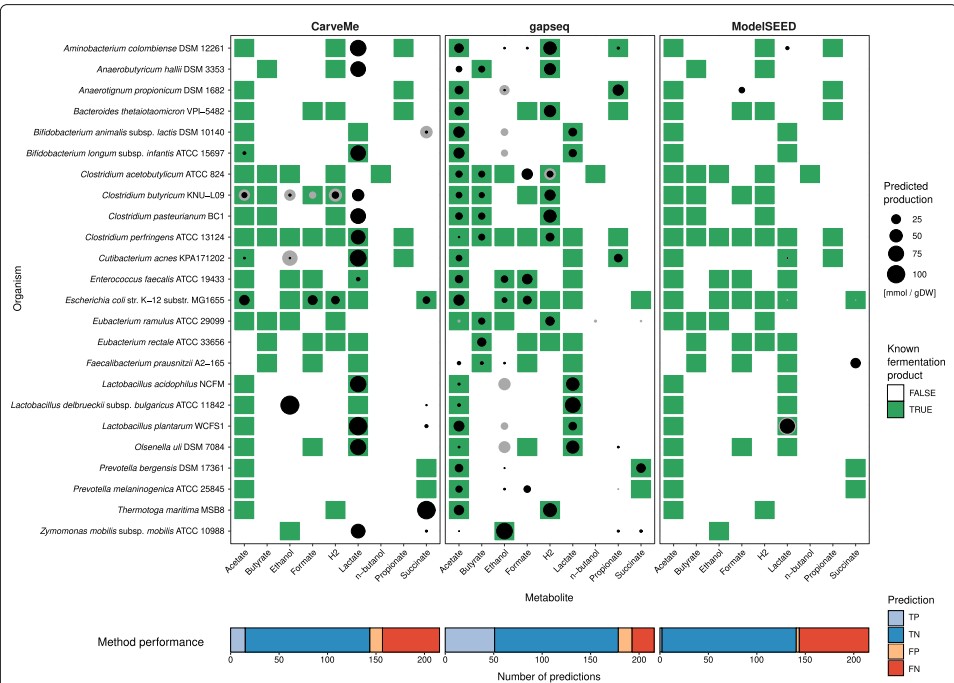

**Fig. 3** Results of the fermentation product test of 24 bacterial organisms under anaerobic growth with models generated using `gapseq`, CarveMe, and ModelSEED. Point sizes indicate the predicted production of a fermentation product metabolite (columns) by the corresponding organism (row). Predictions (black) are based on Minimise-Total-Flux (MTF) flux balance analyses. Grey circles indicate the upper production limit obtained from Flux-Variability-Analysis (FVA). Metabolite-organism-combinations highlighted in green denote known fermentation products, which have been reported in literature based on experimental measures of the metabolite in anaerobic cultures

## Anaerobic food web of the gut microbiome

The prediction of metabolic interactions between microbial organisms is of special interest in ecology, medicine, and biotechnology. So far, we showed the capacity of `gapseq` on the level of individual models. In a next step, we simulated several individual models together as a multi-species community to validate the potential of `gapseq` in microbial community modelling. As sample application we selected representative members of the gut microbiome that are known to form an anaerobic food web [50, 51]. Altogether, we employed 20 organisms and simulated the combined growth in a shared environment for several time steps using the community modelling framework BacArena [52]. BacArena permits a dynamic and spatial simulation of individual models which are optimised separately in a shared growth environment. Based on metabolic models and environmental substance availability, BacArena predicts growth and nutrient exchanges of individual microorganisms and overall alteration in substance concentrations. Metabolite production and consumption rates by individual community members was analysed at time step 3 for CarveMe and `gapseq` and at time step 5 for ModelSEED, to ensure the community metabolism is captured during the exponential growth phase before the inflection point (Fig. 4).

On the community level, simulations using `gapseq` models captured the central substances, which are known to be produced in the context of the food web (Fig. 4). This included the production of short-chain fatty acids (acetate, propionate, butyrate), lactate,

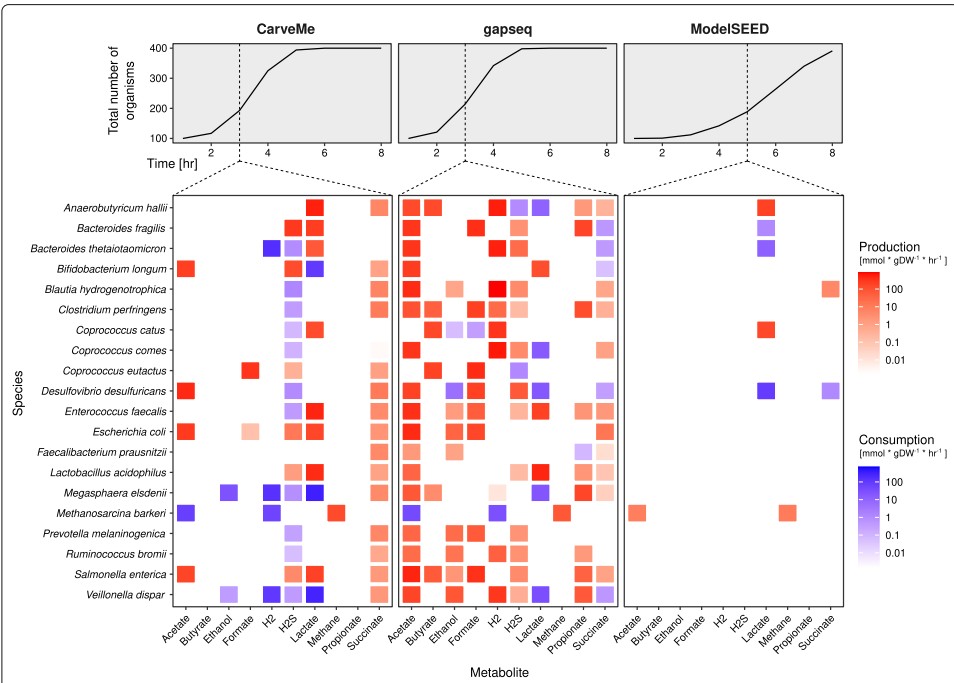

**Fig. 4** Predicted metabolic products and food web of an anaerobic microbial community. The metabolism of a community consisting of 19 bacterial species commonly found in the human gut and one archaeon (*Methanosarcina barkeri*) was predicted using BacArena [52]. Growth curves show the sum of organisms in the shared environment over the simulation time. The heatmaps display the predicted metabolite production (red) and consumption (blue) during exponential growth before the inflection point. Production and consumption rates at all time steps are shown in Additional file 1, Fig. S5. All bacterial models were reconstructed by CarveMe, `gapseq`, or ModelSEED, with the exception of *M. barkeri* for which a published and manually curated model [117] was used

hydrogen, hydrogen sulphide ($H_2S$), methane, ethanol, formate, and succinate. The formation of acetate, formate, and hydrogen was most prevalent, which are also common end products of intestinal fermentation. With the exception of butyrate and methane, parts of the produced fermentation products are further metabolised by some community members (Fig. 4). The predicted identity of fermentation end products and other by-products of metabolism was found in most cases to be closely in line with literature information [50, 51, 53]. For example, the formation of lactate was observed in the simulation for *Lactobacillus acidophilus*, *Enterococcus faecalis*, and *Bifidobacterium longum*, and butyrate was released by known butyrate producers, including *Anaerobutyricum hallii*, *Clostridium perfringens*, *Coprococcus* spp., and *Megasphaera elsdenii*. Yet, the predictions did not include known butyrate production by *Faecalibacterium prausnitzii*. In general, the main products of mixed acid fermentation (acetate, formate, hydrogen, ethanol, succinate) were predicted for diverse members of the community which is in agreement with what is known about common metabolic end products of many gut-dwelling microorganisms [53]. Specifically, high levels of $H_2$ production was correctly predicted for known hydrogen producers including *A. hallii*, *Bacteroides thetaiotaomicron*, *Coprococcus catus*, *Coprococcus comes*, and *Veillonella dispar*.

In general, the anaerobic oxidation of fatty acids is not favoured by the gut environment because the host competes for the uptake of butyrate, propionate, and acetate, which serve as energy source for colonic epithelial cells and are involved in many host functions [54].

Therefore, the gut community lacks syntrophic organisms which are able to anaerobically degrade butyrate [55]. In agreement with this, we found no microbial uptake of butyrate in the community simulation. In contrast, cross-feeding interactions that involve the uptake of metabolites such as acetate, lactate, succinate, and hydrogen are important components in the microbial ecology within the large intestine of humans [56–58]. In our simulations, lactate was predicted to be produced and consumed by distinct community members. We found utilisation of lactate by *A. hallii*, *C. comes*, *Desulfovibrio desulfuricans*, *M. elsdenii*, and *V. dispar*, which is a known feature of these organisms [50, 59]. In addition, succinate was correctly predicted to be utilised by *Bacteroides* species [53]. The formation of methane is known to be limited to methanogenic archaea, and thus *Methanosarcina barkeri* produced methane from acetate and hydrogen during our simulations. It also needs to be noted that certain known cross-feeding interactions were not observed in the community simulations. *A. hallii* and *F. prausnitzii* have been described to consume acetate that is produced by other community members, yet, this cross-feeding is not part of the predicted food web (Fig. 4). Also, no utilisation of gut bacteria-derived hydrogen by *Blautia hydrogenotrophica* as source of energy [60] was predicted. In order to investigate the causes of missing metabolite consumption predictions, the uptake fluxes of *A. hallii*, *F. prausnitzii*, and *B. hydrogenotrophica* were analysed. All three organisms utilised saccharides (i.e. glucose and fructose) as main sources of energy instead of acetate (*A. hallii* and *F. prausnitzii*) or $H_2$ (*B. hydrogenotrophica*). This suggests, that the correct prediction of the anaerobic utilisation of low energy-yielding substrates, such as acetate, remains a challenge for automatic model reconstructions. Specifically acetate was also identified in the carbon source test (see Carbon source utilisation), whose utilisation predictions failed to recapitulate reported acetate utilisation properties of bacteria in nearly half of the cases.

For comparison, the community simulations were also performed using models reconstructed with CarveMe and ModelSEED (Fig. 4 and Additional file 1: Fig. S2). In both cases, most of the above-mentioned known metabolic cross-feeding interactions and end products were not predicted. For instance the production of the short-chain fatty acids butyrate and propionate was missing. The expected consumption of $H_2$ by *B. hydrogenotrophica* and acetate by *A. hallii* and *F. prausnitzii*, which were not predicted in the community simulations using gapseq models were also missing in the simulations with CarveMe and ModelSEED reconstructions.

In summary, gapseq models were able to recapitulate pivotal interactions, which are described for microbial communities in the human gut. While not all expected cross-feeding interactions were recapitulated in the community simulation, important individual contributions to the production and consumption of microbial metabolites in an anaerobic environment were predominantly found to be in close agreement with literature data. Taken together, the community simulation results illustrate the capacity of gapseq to construct predictive models for complex metabolic interaction networks comprising several different species.

### Pathway prediction of soil and gut microorganisms

To demonstrate the pathway prediction capabilities of gapseq, we analysed two communities of soil and gut microorganisms comprising 922 and 822 organisms, repectively. The two communities could be separated from each other by differences in energy metabolism (principal component analysis, Fig. 5a). Here, most variance was explained by subsystems

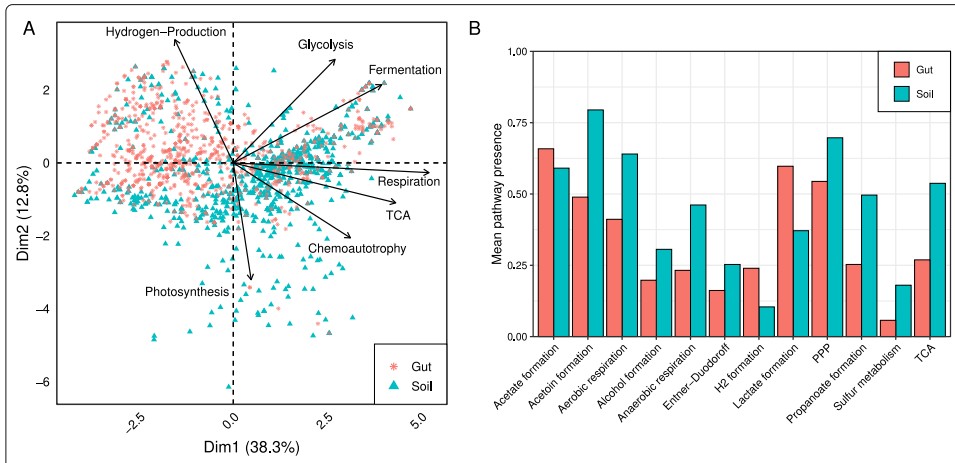

**Fig. 5** Comparison of energy metabolism between soil and gut community. **a** A PCA plot with the first two dimension explaining more than 50% of the variance. Selection of subsystems from energy metabolism with highest quality and impact are shown. **b** List of subsystems of energy metabolism that differ significantly in frequency between members of the soil and gut community (TCA, tricarboxylic acid cycle; PPP, Pentose phosphate pathway)

of pathways that are involved in chemoautotrophic, respiratory, and fermentative processes including hydrogen production. Out of 128 energy pathways, the presence of 40 pathways differed significantly (Kolmogorov-Smirnov test, $P < 0.05$) between soil and gut microorganisms and could be categorised into 12 subsystems (Fig. 5b). In total, gut microorganisms showed less variety in energy pathways than soil microorganisms. Only pathways relevant for the formation of acetate, hydrogen, and lactate were predicted to be enriched. In the case of all other energy subsystems, more pathways were predicted for soil organisms, most prominently pathways relevant for aerobic and anaerobic respiration as well as the tricarboxylic acid cycle (TCA). In summary, members of the soil community showed a more versatile energy metabolisms, which potentially indicates a higher energetic specialisation of gut microorganisms. This sample application demonstrates how `gapseq` can facilitate the characterisation and comparison of microbial communities based on the analysis of the presence and absence of specific metabolic pathways.

### Model reconstructions for metagenomic assemblies

Genome-scale metabolic models can also be reconstructed on the basis of species-level genome bins (SGBs, [61]) assembled from shotgun metagenomic sequencing reads. Yet, genome assemblies from metagenomic material are more prone to errors, fragmentation, and sequence gaps than assemblies of isolated genomes [62], which can potentially cause gaps in the metabolic network reconstructions. We tested whether `gapseq` is able to identify and fill such gaps by comparing the models reconstructed for 127 SGBs from the human microbiome [61] to corresponding models of closely related reference genomes that were assembled from DNA-sequencing of pure cultures (Additional file 1: Fig. S3).

As expected, we found a strong positive correlation between the SGBs' genome completion and their model similarity to their respective reference models (Spearman's rank correlation, n = 127, $P < 10^{-9}$). To estimate the quantitative effect of genome completion on the model similarity, a logarithmic function ($y(x) = c + b * \log(x)$) was fitted to

the data ($R^2 = 0.71$, Additional file 1: Fig. S3). The fitted model indicated, that gapseq is able to reconstruct the underlying metabolic network of an organism even on the basis of incomplete and fragmented genomes. For instance, `gapseq` was on average able to recover 90% of the enzymatic reactions that are found in the reference models for SGBs with a predicted genome completion of only 80% (Additional file 1, Fig. S3).

### Summary of validation tests

In summary, gapseq was evaluated on the basis of five validation tests: (1) The predictions of specific enzymes were compared to experimental data of enzyme activities for a wide range of bacterial strains. The experimental data was retrieved from the Bac-Dive database [45]. (2) The ability of bacterial metabolic models to utilise certain carbon sources was scrutinised by comparing predicted utilisation with data from ProTraits [46], a resource of 424 literature- and genome-inferred prokaryotic phenotypes for more than 3000 organisms. (3) Predicted essentiality of genes was evaluated on the basis of in silico gene-knockout simulations and empirical essentiality data from single gene-knockout studies spanning five bacterial strains. (4) Predicted fermentation products of 24 bacteria in an anaerobic environment were contrasted with fermentation end products reported in scientific literature. (5) An anaerobic microbial community was simulated with reconstructed metabolic models in a shared in silico growth environment. Predicted metabolite production and consumption was compared to those reported in scientific literature.

The overall accuracy (proportion of all correct prediction in relation to all predictions made) of model predictions with empirical data was 66% (CarveMe), 70% (ModelSEED), and 81% (`gapseq`)(Table 1). Sensitivity measures the proportion of correctly predicted positives, whereas specificity accounts for the correct prediction of negatives. All approaches showed a high specificity >0.7 with highest values for fermentation product and gene essentiality tests. Notably, `gapseq` showed the highest sensitivity over all tests (Fig. 6). In summary, `gapseq` outperformed other methods in terms of accuracy and sensitivity while showing similar specificity.

**Table 1** Summarised comparison of CarveMe, gapseq, and ModelSEED

| Metric | CarveMe | gapseq | ModelSEED |
|---|---|---|---|
| *Implementation* | | | |
| Infrastructure | Local | Local | Web service |
| Input (FASTA file) | Protein | Nucleotide | Nucleotide |
| Programming languages | Python | Shell script, R | Perl/javascript |
| Gap-fill solver | CPLEX | GLPK/CPLEX | Not needed* |
| Gap-fill problem formulation | MILP | LP | MILP |
| | | | |
| *Performance* | | | |
| Accuracy | 0.66 | 0.80 | 0.69 |
| Sensitivity | 0.34 | 0.71 | 0.33 |
| Specificity | 0.85 | 0.82 | 0.88 |
| Model file quality** | 0.32 ± 0.006 | 0.78 ± 0.004 | 0.39 ± 0.016 |

Accuracy, sensitivity, and specificity scores are based on 14,931 tested phenotypes including energy sources, enzyme activity, fermentation products, gene essentiality, and anaerobic food web structure predictions.
*Solver runs on ModelSEED server. No local solver is required.
**MEMOTE total score mean (± SD).

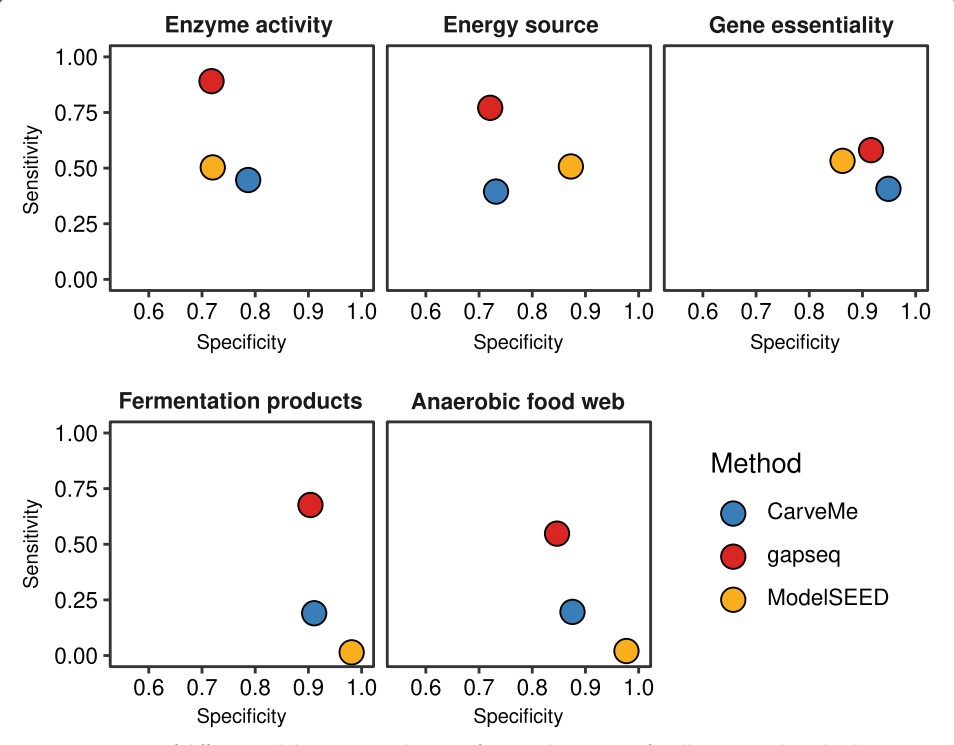

**Fig. 6** Summary of different validation tests. The specificity and sensitivity for all compared methods are shown. This includes results from benchmarks concerning enzyme activities, energy sources, fermentation products, gene essentiality, and metabolite production/consumption in an anaerobic food web

## Comparison with current genome-scale metabolic network reconstruction tools

The above benchmark tests compared `gapseq` with two other tools (CarveMe and ModelSEED) which are able to reconstruct models that enable FBA-simulations of cell growth to predict reaction activity. In this subsection and on the basis of key criteria defined by Mendoza et al. (2019) [28] for the assessment of reconstruction software, `gapseq` is compared to a broader range of currently available network reconstruction tools (Table 2).

**Table 2** Comparison of gapseq (GS) with other reconstruction tools based on criteria defined by Mendoza et al., 2019 [28]

| Feature/comparison criterion | AU | CM | ME | MD | MS | PT | RA | GS |
|---|---|---|---|---|---|---|---|---|
| Software maintenance/support/updates | ● | ● | ● | ● | ● | ● | ● | ● |
| Eukaryotes model support | ● | ○ | ● | ● | ● | ● | ● | |
| SBML level 3 as output | ● | ● | ● | ● | | | ● | ● |
| User-friendly interface | ○ | ○ | ● | ● | ● | ● | ○ | ○ |
| Open Source (source code is open to all users) | ● | ● | | ● | ● | | ● | ● |
| Automatisation until FBA-functional models* | ○ | ● | | ○ | ● | ○ | ○ | ● |
| Manual refinement assistance | ○ | | ● | | | ● | ○ | ○ |
| Customisable for a high number of genomes | | ● | | | ● | ○ | ● | ● |
| Traceability | ○ | | ○ | | | ○ | | ● |
| Automatic refinement using experimental data | ○ | | | | | ○ | | |

Evaluations for AuReMe (AU, [29]), CarveMe (CM, [30]), merlin (ME, [31]), MetaDraft (MD, [32]), ModelSEED (MS, [33]), Pathway Tools (PT, [34]), and RAVEN (RA, [35]) are directly adopted from Mendoza et al., (2019) [28], with the exception of the SBML output, where methods were only classified based on whether the export in SBML level 3 is supported. Legend: ● - outstanding; ○ - good to satisfactory; no circle - poor to unsatisfactory
*Models with FBA-predicted flux through biomass reactions on a given growth medium

The comparison aims to aid potential users to decide when to use `gapseq` and when other tools might be more fitting to their specific research question.

As all other tools listed in Table 2, `gapseq` is maintained by a core team of scientists that provides updates and user support. Issues can directly be reported and additional features requested at the github repository [63], where also the latest version of the software (incl. its source code) can be obtained. `gapseq` exports models in standard SBML level 3 format [64], which enables the integration of `gapseq` in pipelines that further analyse the models with other tools for constraint-based analysis. Additionally, gapseq stores models as R-objects of class *modelorg*, which can be analysed in R using the *sybil* package [65]. As mentioned above, `gapseq` enables the full automatisation of the reconstruction process from the genome to a FBA-functional model that allows growth predictions for the focal organism under a given growth environment. This feature is only shared with CarveMe and ModelSEED and is especially relevant in situations where large numbers of genomes are subject to genome-scale network reconstruction and directly subsequent metabolic flux simulations of microbial growth. Another advantage of `gapseq` is the high traceability of reactions and metabolites throughout the entire reconstruction process: In the final model, `gapseq` adds a flag to each reaction that denotes why the specific reactions have been added to the network, e.g. due to sequence homology to reference proteins, or at which gap-filling step. This information is stored in the reactions' attributes within the model's R-object and could be highly relevant for further manual refinement of the network model.

For certain metabolic network reconstruction projects, other tools than `gapseq` might be more fitting: (i) `gapseq` does not support the construction of genome-scale models for eukaryotic organisms. The tools AuReMe, merlin, MetaDraft, ModelSEED, Pathway Tools, and RAVEN explicitly provide this feature (Table 2). (ii) `gapseq` does not offer a graphical user interface, which might be a hurdle for users less accustomed to command line software tools. (iii) In its current version, `gapseq` does provide a function (`./gapseq adapt`) that allows users to manually add or remove reactions or pathways from a reconstructed network model. However, Pathway Tools and merlin offer workspaces with extended functions and assistance for manual refinement, including network visualisation. (iv) Finally, options to automatically refine models based on experimental data is not yet implemented in `gapseq`, while the tools AuReMe and Pathway Tools provide this feature for certain data types.

## Discussion

Here, we introduced `gapseq`—a new tool for metabolic pathway analysis and genome-scale metabolic network reconstruction. The novelty of `gapseq` lies in the combination of (i) a novel reaction prediction that is based both on genomic sequence homology as well as pathway topology, (ii) a profound curation of the reaction and transporter database to prevent thermodynamically infeasible reaction cycles, and (iii) a reaction evidence score-oriented gap-filling algorithm. In order to scrutinise `gapseq` metabolic models, we compared the models' network structures and predictions with large-scale experimental data sets, which were retrieved from publicly available databases. Furthermore, the ability of `gapseq` to predict bacterial phenotypes was compared to two other commonly used automatic reconstruction methods, namely, CarveMe [30] and ModelSEED [33] (Table 1). ModelSEED is also implemented in the KBASE online software platform [66].

### Crucial large-scale benchmarking of metabolic models

The quality of genome-scale metabolic networks can be assessed by comparing model predictions with experimental physiological data. The protocol by Thiele and Palsson (2010) for the reconstruction of genome-scale metabolic networks recommends the quality assessment and manual network curation using data for (i) known secretion products (e.g. fermentation end products), (ii) single gene deletion mutant growth phenotypes (i.e. gene essentiality), and (iii) the utilisation of carbon/energy sources [22]. Tools for the automatic reconstruction of metabolic networks should also make use of such physiological data whenever available for benchmarking. Here, we tested our gapseq approach on the basis of all three recommended phenotypic data and compared the performance with CarveMe and ModelSEED. Additionally, we included two novel benchmark tests: The comparison of model predictions with (iv) the activity of specific enzymes known from experimental studies [45] and (v) metabolic interactions among microorganisms in a multi-species community within an anaerobic environment (food web). Across all five benchmark tests, we could show that gapseq outperformed CarveMe and ModelSEED in terms of sensitivity while achieving specificity scores that are comparable to the other two tools (Fig. 6).

Publicly available genome sequences of microorganisms, which can be subject for automated metabolic network reconstruction are massively increasing in number due to continuing advances in high-quality and high-throughput sequencing technologies [21]. This development is further fuelled by the increasing number of genome assemblies from metagenomic material [67]. In contrast, standardised phenotypic data for microorganisms remains a bottleneck for the validation of automated metabolic network reconstruction pipelines such as gapseq. As consequence, it is crucial for the future development of automated network reconstruction software to include possibly all available phenotypic data for benchmarking, especially data from non-model organisms. To benchmark gapseq in relation to CarveMe and ModelSEED using phenotypic data from mainly non-model organisms, we retrieved phenotypic data of enzyme activity for more than 3000 organisms and carbon source utilisation for more than 500 organisms from online databases, which is, to our knowledge, the yet largest phenotypic data set used for validation of automatically reconstructed metabolic networks. In this validation approach gapseq achieved the highest prediction accuracy among all three tools tested (Fig. 1, Table 1).

Hence, those results suggest that gapseq is a powerful new tool for the automated reconstruction of genome-scale metabolic network models. Moreover, the underlying reference protein sequences as well as the pathway database can readily be updated using online resources, which makes gapseq flexible to include future developments and findings in microbial metabolic physiology.

### Automated network reconstructions for community modelling

While single organisms can be considered as the building blocks of microbial communities, individual metabolic models of organisms are the building blocks of in silico microbial community simulations. Therefore, genome-scale metabolic models are increasingly applied to predict the function of multi-species microbial communities [68–70]. To correctly infer metabolic interaction networks between different organisms, it is important that individual models accurately predict nutrient utilisation (e.g. carbon

source) and metabolic end products (e.g. fermentation products). In this study, the benchmarks for carbon source utilisation and fermentation end product identity indicated that `gapseq` has the highest prediction performance compared to other reconstruction tools (Figs. 1 and 3).

To illustrate the applicability of `gapseq`-reconstructed metabolic models for the simulation of multi-species community metabolism, we generated models for microbial strains from the gut microbiota and simulated their growth in a shared environment. Without further curation, the community simulation reproduced important hallmarks of intestinal anaerobic food webs [50, 53]. Above all, short-chain fatty acids (SCFA) were predicted to be the primary end products of fermentation. This prediction is important to represent intestinal metabolism, because SCFA are crucially involved in host physiology by affecting regulatory response in intestinal and immune cells [71, 72]. Furthermore, the simulation accurately predicted the exchange of metabolites between different members of the microbial community (Fig. 4). Cross-feeding of metabolites and the formation of anaerobic food chains have been associated with a healthy microbiome [11, 73]. For instance, the cross-feeding of lactate has been reported to be vital for the early establishment of a healthy gut microbiota in infants [73]. Accordingly we observed the exchange of lactate between different bacterial species in the community simulations (Fig. 4) and involved known lactate producers (e.g. *Enterococcus faecalis*) and consumers (e.g. *Megasphaera elsdenii*). This example illustrates that we are able to predict key features of the anaerobic food web within the gastrointestinal microbiota using `gapseq` models. In addition to the ability to accurately model metabolic processes within existing microbial communities, `gapseq` will further promote the potential of metabolic modelling to predict how complex microbial communities can be modulated by targeted interventions. Specific interventions, which could for instance be predicted, are the introduction of new species to the community (i.e. probiotics) or microbiome-modulating compounds (prebiotics) to the environment. Predictions of potential intervention strategies which target the microbiome are of vast relevance for biomedical research. Furthermore, metabolic interactions between microbiome members are difficult to detect in vivo due to the simultaneous production and uptake of metabolites. Thus, in silico predictions of metabolite cross-feeding interactions are highly valuable for hypothesis generation about the function and dynamics of microbial communities.

Taken together, the results obtained with `gapseq` suggest, that metabolic models which are reconstructed using `gapseq` are promising starting points to construct ecosystem-scale models of inter-species biochemical processes and to predict targeted strategies to modulate microbiome structure and function.

### Pathway analysis of microbial communities

The construction of genome-scale metabolic models is based on metabolic networks that are inferred from genomic sequences in the context of biochemical databases [22]. Although the reconstruction of metabolic networks is closely related to the prediction of metabolic pathways, metabolic modelling and pathway analysis are often treated separately [74]. In `gapseq`, the prediction of metabolic pathways is intrinsically tied to the reconstruction of metabolic networks and gap-filling. In addition, reaction, transporter, and pathway predictions can also be used to evaluate the functional capacities of microorganisms without the need of metabolic modelling. As an example for metabolic pathway

analysis, we compared the predicted energy metabolism of two large microbial communities that occur in soil and the human gut. We could show that the predicted distribution of pathways differ between both communities based on the habitat, which usually accommodates the members of the respective community. Gut microorganisms showed a less versatile energy metabolism and a specialisation towards fermentation pathways, which lead to the formation of acetate, hydrogen, and lactate. Variations in pathways distributions between both communities may be explained by distinct evolutionary histories. The habitat of the diverse group of soil microorganisms more likely represents an open ecosystem, whereas the gut microbiome is directly constraint by a multi-cellular host that potentially affect microbial phenotypic traits [75]. In general, metabolic modelling should be accompanied by the analysis of pathways based on statistical methods [74] to compensate for additional assumptions, which are introduced in constraint-based metabolic flux modelling [4].

### Limitations and outlook

`gapseq` requires 1–2 h for the reconstruction of a single model, whereas ModelSEED and CarveMe operate faster (10 min) on a standard desktop computer. Nonetheless, CarveMe needs as input gene sequences (protein or nucleotide), which has to be predicted first, and ModelSEED works as a web service, which can complicate the handling of large-scale reconstruction projects. In `gapseq`, pathways were predicted based on topology and sequence homology searches. However, the assignment of enzymatic function from sequence comparisons has been shown to potentially miss protein domain structures and thus can cause false annotations [76, 77]. In addition, `gapseq` employs many resources to find potential sequences for reactions in pathway databases. Together this might explain why although `gapseq` performed better than other methods on predicting positive phenotypes (function present), it went head to head with regard to negative phenotype predictions (function not present). CarveMe takes a different approach when inferring function by taking care of functional regions (protein domains) [78], resulting in orthologous groups [79], which results in a slightly better specificity (true negative phenotype predictions) in benchmarks (Fig. 6). Future developments of `gapseq` will address orthologous groups by using multiple inference methods. The integration of functional predictions coming from phylogenetic inference without the need of genomic sequences [80] might also be promising for further developments of `gapseq`. Moreover, future versions of `gapseq` will address challenges that we identified in the course of the evaluation tests presented in this study. For instance, Gene-Protein-Reaction (GPR) association predictions will be improved by incorporating new computational methods in protein complex detection [81].

### Conclusion

We provide a new software tool called `gapseq` that is suitable for metabolic network analysis and metabolic model reconstruction. To enhance phenotype predictions, `gapseq` employs various data sources and a novel gap-filling procedure that reduces the impact of arbitrary growth medium requirements. We further brought together the so far largest benchmarking of genome-scale metabolic models, in which `gapseq` outperformed comparable alternative tools. With the increased model quality of automated network reconstructions, `gapseq` will provide new insights into the metabolic

phenotypes of non-model and yet-uncultured bacteria whose genomes are assembled from metagenomic material. In this way, the models and their simulations allow predictions on the organisms' ecological role in their natural environments. Taken together, we consider gapseq as important contribution to the modelling of microbial communities in the age of the microbiome.

## Methods

### Program overview and source code availability

The source code is accessible and maintained at https://github.com/jotech/gapseq. The program is called by ./gapseq, which is a wrapper script for the main modules. Important program calls are ./gapseq find (pathway and reaction finder), ./gapseq find-transport (transporter detection), ./gapseq draft (draft model generation), ./gapseq fill (gap-filling), or ./gapseq doall to perform all in line. When ever necessary, method sections directly refer to config, data and source code files from the gapseq package, which contains the main sub-directory src/ with source code files and dat/, which contains databases and also the sequence files in dat/seq/. Figure 7 shows an overview of the different gapseq modules. Documentation and tutorials for gapseq can be found at https://gapseq.readthedocs.io.

### Pathway, subsystem, and sequence databases

It is crucial to link pathways and subsystems to protein sequences of the involved enzymes, which can be employed for homology search. Pathways are considered as a list of reactions with enzyme names and EC numbers. In addition, pathways can be generalised as subsystems, which are sets of 'functional roles that together implement a specific biological process' [82]. Pathway and subsystem definitions were obtained from Meta-Cyc [26], KEGG [24], and ModelSEED [33]. For MetaCyc, PathwayTools [34] was used in combination with PythonCyc to obtain pathway definitions [83] (src/meta2pwy.py). Information on Kegg pathways were retrieved directly from the KEGG homepage: reactions (http://rest.kegg.jp/list/reaction), and EC numbers (http://rest.kegg.jp/link/

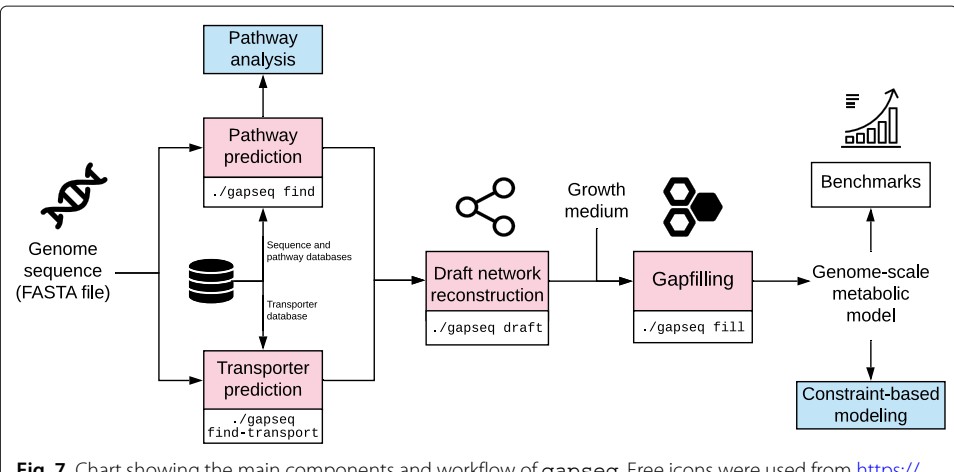

**Fig. 7** Chart showing the main components and workflow of gapseq. Free icons were used from https://www.flaticon.com (creators: Freepik, Gregor Cresnar, Freepik, Smashicons)

pathway/ec) and further processed (src/kegg_pwy.R). In case of ModelSEED, subsystem definition were obtained from the homepage: http://modelseed.org/genomes/Annotations (src/seed_pwy.R). In addition, manual defined and revised pathways are stored in the file dat/custom_pwy.tbl.

Amino acid sequence (protein) data required for pathway prediction were retrieved from UniProt [84] for each reaction identified by EC number, enzyme name, or cross-references (curated UniProt IDs stored in other databases). Both reviewed and unreviewed sequences are considered and stored as clustered UniPac sequences (src/uniprot.sh). To increase the sequence pool for a given reaction, alternative EC numbers from BRENDA [85] and from the Enzyme Nomenclature Committee https://www.qmul.ac.uk/sbcs/iubmb/enzyme/ are integrated (src/altec.R, dat/brenda_ec.csv). For the download from UniProt, EC numbers and database cross-references are prioritised over enzyme names because the matching is often ambiguous. For a default gapseq run, 95% of the reactions have associated EC numbers and for 75% of the reactions without EC number cross-references to UniProt IDs are available. From the available EC numbers, 66% are specific (i.e. have a full four-level number code) and cross-references to UniProt IDs exist for 86% of the unspecific EC numbers. In 1.8% of cases, multiple EC numbers belong to one reaction. In those cases, each EC number is considered as an individual reaction.

### Pathway and subsystem prediction

As a first step, gapseq predicts the presence of metabolic pathways based on the organism's genome sequence. For each pathway or subsystem selected from a pathway database (MetaCyc, KEGG, ModelSEED, custom), gapseq searches for sequence evidence and a pathway or subsystem is defined as present if enough of its reactions were found to have sequence evidence. In more detail, sequence data (see "Pathway, subsystem, and sequence databases" subsection) is used for homology search by *tblastn* [86] with the protein sequence as query and the genome as database. By default, a bitscore $\geq$ 200 and a coverage of at least 75% is needed for a match. For certain reactions, the user can define additional criteria, for example an identity of $\geq$ 75% (dat/exception.tbl). In case of protein complexes with subunits, a more complex procedure is followed ('Protein complex prediction' section). Spontaneous reactions, which do not need an enzyme, were set to be present in any case. In general, a pathway or subsystem is considered to be present if at least 80% of the reactions are found (completenessCutoffNoHints threshold). This completeness threshold is lowered for pathways or subsystems in following cases:

1  If the pathway or subsystem contains key reactions, as it is defined for a large number of pathways in MetaCyc, and all key reactions are found, then completenessCutoff of the total reactions needed to be found. We used a value of 2/3 for this threshold.

2  In cases in which no sequence data is available for specific reactions, the status of the reactions is set to 'vague' and these reactions do not count as missing if they account for less than vagueCutoff of the total reactions of a pathway or subsystem. We used a value of 1/3 for this threshold.

It is important to note that gapseq uses MetaCyc's *base pathway* as default pathway structures and we highly recommend to use this default setting for genome-scale

metabolic model reconstructions. This is because MetaCyc provides *base pathway* definitions that follow strict criteria: First, these pathways represent experimentally determined metabolic routes for small molecules (metabolites), which are curated on the basis of scientific literature [87]. Second, the *base pathway* definition includes that these pathways are composed of reactions only, 'where no portion of the pathway is designated as a subpathway' [88]. These criteria results in a larger number of pathways of smaller size compared to KEGG pathways or ModelSEED subsystems. It was previously emphasised that the smaller pathway structures of MetaCyc allow more focused predictions of pathway existence from sequenced genomes [89]. Alternatively, users may choose to use KEGG pathways or ModelSEED subsystems for pathway predictions. This option could be of interest for users, who intend to use `gapseq` solely to investigate enzyme presence and pathway/subsystem coverage from a genome on the basis of reference pathways other than MetaCyc pathways.

The pathway and subsystem prediction algorithm is implemented in the bash shell script `src/gapseq_find.sh`, which uses GNU parallel [90] and fastaindex/fastafetch from exonerate [91].

### Protein complex prediction

A problem with automatic sequence download for reaction-associated reference proteins (as FASTA files) comes with protein complexes, for which a single blast hit may be not sufficient to predict enzyme presence. In `gapseq`, the subunit information of protein complex components is extracted from the sequence FASTA headers of the reaction-associated protein sequences obtained as references from UniProt. Search terms are: 'subunit', 'chain', 'polypeptide', 'component', and different numbering systems (roman, arabic, greek) are homogenised. To avoid artefacts in text matching, subunits that occur less than five times in the sequence file are not considered, and in cases in which a subunit occurs almost exclusively ($\geq$ 66%) the other entries are not taken into account. All FASTA entries, which could not be matched by text mining, or which were excluded because of the coverage, are labelled 'undefined subunit' and do not add to the total amount of subunits. For each recognised subunit, a blast search is performed. A protein complex is considered present if more than 50% of the subunits could be found, whereby the presence of 'undefined subunits' tip the balance if exactly 50% of the subunits were found. The text matching with regular expressions is done with R's stringr [92] and Biostrings [93] as defined in `src/complex_detection.R`. The script is called from within the shell script `src/gapseq_find.sh`.

### Transporter prediction

Transport reactions govern the exchange of metabolites with the environment and are therefore essential for phenotype predictions. For transporter search, sequence data from the Transporter Classification Database (TCDB) is employed [94]. In addition, manual defined sequences can be defined in `dat/seq/transporter.fasta`. The sequence set is reduced to a subset of transporters that involve metabolites known to be produced or consumed by microorganisms (`dat/sub2pwy.csv`). Subsequently, the genome is queried by the reduced sequences using *tblastn* [86]. For each hit (default cutoffs: bitscore $\geq$ 200 and coverage $\geq$ 75%), the transporter type (1. Channels and pores, 2. Electrochemical potential-driven transporter, 3. Primary active transporters, 4. Group translocators) is

determined using the TC number mentioned in the FASTA header of the source sequence from TCDB. A suitable candidate reaction is searched in the reaction database. If there is a hit for a transporter of a substance but no candidate reaction for the respective transporter type can be found, then other transporter types are considered. The transporter search is done by the shell script `src/transporter.sh` that uses GNU parallel [90] and fastaindex/fastafetch from exonerate [91].

Candidate transporters are selected from the reaction database by transporter type and substance name. This is done by text search and is currently implemented only for the ModelSEED namespace. From the ModelSEED reaction database all reaction with the flag *is_transport* = 1 are taken into account and the transporter type is predicted by keywords: 'channel', 'pore' (1. Channels and pores); 'uniport', 'symport', 'antiport', 'permease', 'gradient' (2. Electrochemical potential-driven transporters); 'ABC', 'ATPase', 'ATP' (3. Primary active transporters); and 'PTS' (4. Group translocators). If no transporter type could be identified by keywords, additional string matching is done for ATPases, proton/sodium antiporter, and PTS by considering the stoichiometry of the involved metabolites. The transported substance is identified as the substance that occurs on both sides of the reaction. In addition, reactions from the reaction database can be linked manually to substances and transporter types (`dat/seed_transporter_custom.tbl`). The text matching with regular expressions is done with stringr [92] (`src/seed_transporter.R`).

### Biochemistry database curation and construction of universal metabolic model

For the construction of genome-scale metabolic network models, `gapseq` uses a reaction and metabolite database that is derived from the ModelSEED database [33] as from January 2018. In addition, 67 new reactions and 12 new metabolites were introduced to the `gapseq` biochemistry database (see Additional file 2: Table S1). All reactions and metabolites from the database were included for the construction of a full universal metabolic network model; an approach that is also used in CarveMe [30]. We curated the underlying biochemistry database in order to correct inconsistencies in reaction stoichiometries and reversibilities. Inconsistencies were identified by optimising the universal network model for ATP-production without any nutritional input to the model using flux balance analysis. In case of ATP-production, the flux distributions of such thermodynamically infeasible reaction cycles were investigated by cross-checking the involved reactions with literature information, the BRENDA database for enzymes [85], and the MetaCyc database [26]. Stoichiometries and reversibilities of erroneous reactions were corrected accordingly. This curation procedure was repeated until no theromodynamically infeasible and ATP-generating reaction cycles were observed. In total, more than 1500 reactions were subject to corrections in their stoichiometry and/or reversibility.

It needs to be noted, that since we have retrieved the biochemistry database from ModelSEED for the development of `gapseq`, the ModelSEED database has been comprehensively improved and extended [95]. During the development of `gapseq` we have transferred corrections made by the ModelSEED community also to our database. The `gapseq`-developer team will continue this process together with future developments of the ModelSEED database and our software.

Hits from the pathway prediction (Pathway and subsystem prediction) and transporter prediction (Transporter prediction) are mapped to the `gapseq` reaction database using

different common identifiers. A majority of reactions are directly matched via their corresponding Enzyme Commission (EC) system identifier [96] and Transporter Classification (TC) system identifier [94], respectively. For this mapping, also alternative EC numbers for enzymatic reactions as defined in the BRENDA database [85] are considered. Moreover, the databases used for pathway and transporter predictions often provide cross-links to the reaction's KEGG ID, which is also assigned to most reactions in the `gapseq` database and used to match reactions. Additionally, the MNXref database [97] provides cross-links between several biochemistry databases, which `gapseq` also utilises to translate hits from the pathway predictions to model reactions. Finally, a manual translation of enzyme names to model reactions is done for some reactions, which we identified as important reactions but which failed to match between the pathway databases (Pathway and subsystem prediction) and the `gapseq` model reactions using other reaction identifiers (`dat/seed_Enzyme_Name_Reactions_Aliases.tsv`). The overall mapping is done by the function `getDBhit()` as defined in `./src/gapseq_find.sh`.

### Model draft generation

A draft genome-scale metabolic model is constructed based on the results from the pathway and transporter predictions (see above). A reaction is added to the draft model if the corresponding enzyme/transporter was directly found (i.e. the blast hit reached the bitscore cutoff value) or if the pathway was predicted to be present (i.e. due to pathway completeness and key enzymes) in which the reaction participates. Additionally, spontaneous reactions as defined in the MetaCyc database as well as transport reaction of compounds, which are know to be able to cross cell membranes by means of diffusion (e.g. $H_2$), are directly added to every draft model. As part of the draft model construction `gapseq` adds a biomass reaction to the network that aims to describe the composition of molecular constituents that the organism needs to produce in order to form 1 g dry weight (1 gDW) of bacterial biomass. `gapseq` uses the biomass composition definition from the ModelSEED database for Gram-positive (`dat/biomass/seed_biomass.DT_gramPos.tsv`) and Gram-negative bacteria (`dat/biomass/seed_biomass.DT_gramNeg.tsv`). If no Gram-staining property is specified by the user, `gapseq` predicts the Gram-staining-dependent biomass reactions by finding the closest 16S-rRNA-gene neighbour using a `blastn` search against reference 16S-rRNA gene sequences from 4647 bacterial species with known Gram-staining properties that are obtained from the ProTraits database [46]. The model draft generation is done by the R script `src/generate_GSdraft.R`.

### Gap-filling algorithm

`gapseq` provides a gap-filling algorithm that adds reactions to the model in order to enable biomass production (i.e. growth) and likely anabolic and catabolic capabilities. The algorithm uses the alignment statistics (i.e. the bitscore) from the pathway- and transporter prediction steps of `gapseq` (see above) to preferentially add reactions to the network, which have the highest genetic evidence. This approach is especially relevant in cases where the sequence similarity to known enzyme-coding reference genes was close to but did not reach the cutoff value *b*, which is required for a reaction to be included directly into the draft network. In contrast to the gap-filling algorithms described in previous works [98] and [30], which also use genetic evidence-weighted gap-filling, the gap-filling

problem in `gapseq` is not formulated as Mixed Integer Linear Program (MILP) but as Linear Program (LP), and is derived from the parsimonious enzyme usage Flux Balance Analysis (pFBA) algorithm developed by Lewis et al., 2010 [3]. Therefore, the alignment statistics (i.e. bitscore) are translated into weights for the corresponding model reactions and incorporated into the problem formulation:

$$
\text{max:} \ \ v_j - c \sum_{i \in R_{\text{all}}} w_i |v_i| \ ,
$$

$$
w_i = \begin{cases} w_{\min} & b_i \geq u \ \ | \ \ i \in R_{\text{draft}} \\ (b_i - u)\left(\frac{w_{\min} - w_{\max}}{u - l}\right) + w_{\min} & l \leq b_i < u \\ w_{\max} & b_i < l \end{cases} \tag{1}
$$

$$
\text{s.t.}
$$

$$
\boldsymbol{S} \cdot \boldsymbol{v} = \boldsymbol{0}
$$

$$
\boldsymbol{lb} \leq \boldsymbol{v} \leq \boldsymbol{ub}
$$

where $R_{\text{all}}$ is the set of all reaction in the universal model, $R_{\text{draft}}$ are the reactions, which are already part of the draft network before gap-filling, $v_j$ is the flux through the objective reactions (e.g. biomass production), $v_i$ the flux through reaction $i$, $w_i$ the weight for reaction $i$, $v$ the flux vector for all reactions, and $c$ a scalar factor that determines the contribution of the absolute reduction of weighted fluxes to the overall FBA solution (default: $c = 0.001$). Moreover, a maximum weight value $w_{\max}$ (default: 100) is assigned if the reaction's highest bitscore is smaller than a threshold $l$ (default: 50). A minimum reaction weight $w_{\min}$ (default: 0.005) is assigned to reactions with a bitscore higher than $u$ (default: 200) or if the reactions are already part of the draft model. $S$ is the stoichiometric matrix and $lb$ and $ub$ the lower and upper flux bound vectors.

Two other LP-based gap-filling algorithms that incorporate reaction evidence scores have been formulated by Dreyfuss et al. (2013) [99] and Medlock et al. (2020) [100], respectively. These approaches require a definition of a minimum flux through the biomass reaction to ensure growth. The pFBA-derived LP formulation of `gapseq` (Eq. 1) includes the flux through the biomass/objective reaction $v_j$ together with the reaction evidence scores in a single objective function.

In `gapseq` and following the solution of the LP (Eq. 1), reactions carrying a flux and which are not part of the draft model are added to the network model. The algorithm is implemented in `src/gapfill4.R`.

### Gap-filling of biomass, carbon sources, and fermentation products

Gap-filling of a draft model in `gapseq` requires only for the first step a user-defined growth medium that is ideally known to support growth of the organism of interest in vivo. If no growth medium is specified by the user, a complete medium (dat/media/ALLmed.csv) is chosen by `gapseq` (as done for the large-scale benchmarks of enzyme activity and carbon sources, cf. Validation with enzymatic data (BacDive), Validation with carbon sources data (ProTraits)). A set of common microbial growth media (e.g. LB, TSB, M9) is provided in the gapseq software directory `dat/medium/`. In addition, the user can provide a custom growth medium definition. The above described gap-filling algorithm is used to improve the generated draft model in four steps. Importantly, steps

2-4 only add reactions having sequence support and aim for improve the model quality without reliance on a specific gap-filling medium.

1  **Biomass production**: To ensure that the model is able to produce biomass under the given nutritional input (gap-filling medium) the gap-filling algorithm is applied while the objective is defined as the flux through the biomass reaction. This step will add all missing reactions that are essential for in silico growth and are not part of the model yet.

2  **Individual biomass components**: It is checked whether the model supports the biosynthesis of individual biomass components. Therefore, the model is re-constrained to a M9-like minimal medium with a carbon source for which an exchange reaction is found (default: glucose if available). The objective function is set to the production of one biomass component at a time and the gap-fill algorithm is performed using only reactions with sequence support as source. This gap-filling step is repeated for each biomass component metabolite twice, with and without oxygen to potentially allow aerobic and anaerobic growth for facultative anaerobes.

3  **Alternative energy sources**: `gapseq` attempts to gap-fill likely metabolic pathways, which enable the utilisation of alternative energy sources, which might not be part of the defined growth medium from step (1). To this end, the model is re-constrained to a M9-like minimal medium containing a single carbon source of interest at the time. Next, three temporary reactions were inserted into the model that recycle common reducing equivalent carriers (ESP1: menaquinol $\rightarrow$ menaquinone $+$ 2H$^+$; ESP2: quinol $\rightarrow$ quinone $+$ 2H$^+$; ESP3: $NADH \rightarrow NAD^+ +$ H$^+$). As objective function, the summed flux of the temporary reactions ESP1, ESP2, and ESP3 is used. Again, the gap-filling of this step only employs those reactions having sequence support. By this, the capacity of a potential carbon source to function as electron donor can be evaluated. This test can be considered as an in silico simulation of the commonly used BIOLOG carbon source utilisation test arrays [47] in which the colometric effect is coupled to a dehydrogenase [101]. This gap-filling step is performed for all metabolites defined in `dat/sub2pwy.csv`.

4  **Metabolic products**: Finally, the same list of compounds (`dat/sub2pwy.csv`), is used to check whether the network can be gap-filled to allow the formation of these metabolites given the original medium. For each compound the gap-filling algorithm is applied with the production of the focal compound as objective function. As for step 2–3, only reactions with sequence evidence are considered for gap-filling.

While step (1) considers all reaction from the universal model as potential candidate reactions for gap-filling, steps (2–4) allow only the addition of candidate reactions to the model with a corresponding bitscore from the pathway prediction (Pathway and subsystem prediction) higher than a threshold value *b* (default: 50). Thus, these so-termed core reactions represent only reactions, for which `gapseq` has found genomic sequence evidence. Gaps in the specific metabolic functions (individual biomass component formation, alternative energy source utilisation, by-product formation) are only resolved if all required additional reactions are core reactions. The gap-filling steps (2–4) are implemented to reduce the impact of a gap-filling medium on the final metabolic model. Thus,

these steps aim to increase the versatility of `gapseq` reconstructions for downstream metabolic simulations of the models in growth environments that are potentially different to the chemical composition of the actual gap-filling medium. This could be especially of relevance for dynamic community metabolism simulations, where the nutritional environment changes over time due to the release of by-products by certain community members that serve as resources for others. In case `gapseq` users prefer to perform gap-filling solely for biomass production on a defined gap-filling medium, the argument `-q` can be passed on to the gap-filling command `./gapseq fill`. The number of reactions added during gap-filling is given as output during runtime. In addition, detailed information on why a specific reaction was added to the model is provided in the reaction attributes table (*@react_attr*) of the model's S4 R-object of class *sybil::modelorg* [65]. In this table, the column *'gs.origin'* states an integer number between 0 and 10 where 0 indicates that the focal reaction was directly included in the draft model due to predicted sequence homology (see Pathway and subsystem prediction and Transporter prediction); 1–4 correspond to the four gap-filling steps as described above; 6 indicates the biomass reaction; 7 and 8 refer to exchange and diffusion reactions, respectively; 9 refers to reactions that were added due to pathway completion; code 10 indicates reactions that are added after using the optional function `gapseq adapt`). The code 5 is currently not used, but might be assigned in a futur e version of `gapseq`.

### Formal and functional model file testing

The validity of genome-scale metabolic model files was checked with MEMOTE (0.10.2) [102]. For all models used in the anaerobic food web (Anaerobic food web of the human gut microbiome), the total MEMOTE score was computed for the respective SBML-Model files. MEMOTE was executed using the parameter `-skip test_find_metabolites_not_produced_with_open_bounds` and `-skip test_find_metabolites_not_consumed_with_open_bounds` since these tests do not contribute to the total MEMOTE score but require long computation time.

### Validation with enzymatic data (BacDive)

Enzyme activity tests are commonly performed for characterisation and identification of microbial isolates. In those tests, microbial cell cultures or extracts from the cultures are exposed to the substrate of the focal enzyme and it is measured whether the substrate is transformed. While the experimental culture and test conditions can vary between microorganisms tested and the specific enzymes of interest, the experiments are generally designed to invoke the expression of the specific enzyme, if the organism harbours the respective gene(s). The Bacterial Diversity Metadatabase (BacDive) is a large curated database for, among other data, results from laboratory enzyme activity tests [45]. We used this information to benchmark automated model reconstructions by scrutinising if the model reconstructions possess the enzymatic reactions, whose activities were tested in laboratory experiments. For this purpose, a list of type strain IDs where downloaded using the advanced search within the BacDive website. Subsequently, the IDs were used to retrieve the strains' data stored at BacDive via the R package BacDiveR (version 0.9.1, [103]). If the stored data contained non-zero entries for enzymatic activity and if a genome assembly was available from NCBI RefSeq, the type strain was considered for the validation analysis (Additional file 2: Table S7). The respective genome assemblies were

downloaded with `ncbi-genome-download` (https://github.com/kblin/ncbi-genome-download). If multiple genomes were available for one type strain, 'representative' and 'complete' (NCBI tags) genomes were preferred and, in case there were still multiple candidate genomes available, the most complete genome was selected. Genome completeness was estimated using the software BUSCO (3.0.2) [104]. In total, 3017 type strain genomes were subject for automated model reconstructions using ModelSEED (2.5.1), CarveMe (1.2.2), and `gapseq`. The gap-filling parameters were set to default values for each program, i.e. a complete medium was assumed. A reaction activity was predicted if the corresponding reaction was found to be present in the model. This was done by matching enzymes and reactions via EC numbers. For CarveMe the vmh (https://www.vmh.life) and for ModelSEED and `gapseq` the ModelSEED (http://modelseed.org) reaction database was used to match reactions and EC numbers. Only those EC numbers were considered for testing for which a matching from EC number to reaction IDs exist. For the EC numbers 3.1.3.1, 3.1.3.2, the corresponding reactions were the same, and thus unspecific, so that both EC numbers were from the validation analysis. In general, the enzymatic data in the BacDive database has the entries *enzyme name*, *EC number*, and experimentally measured *enzyme activity* (active: '+'; not active: '-') but some entries were ambiguous (e.g. '+/-'). These ambiguous entries were omitted from the analysis. If an enzyme was measured to be active according to the BacDive database and the corresponding reaction also present in the metabolic network, then the enzymatic test was called a *true positive*. In contrast, if the reaction was not present in the network the test was called *false negative* (vice versa for false positive, true negative). Sampling of enzymatic data was performed in order to preclude a potential bias due to over-representation of certain EC numbers. All EC numbers with at least 100 tests were considered for sampling. For each EC number, 100 tests were randomly chosen. The re-sampling was repeated 500 times to estimate the variation of true positives, true negatives, false positives, and false negatives.

### Validation with carbon sources data (ProTraits)

In order to predict accurate growth phenotypes using genome-scale metabolic models, it is crucial that the network reconstructions possess the metabolic capability to utilise the specific carbon sources, which the organism can also use in their natural environment. Here, we used microbial phenotypic trait data for the validation of carbon source utilisation from the 'atlas of prokaryotic traits' database (ProTraits) [46]. In brief, ProTraits is an online resource that provides phenotypic trait data spanning over 3000 microorganisms. The data stored in ProTraits represent phenotypes that are inferred from scientific literature and comparative genomics [46]. Each phenotype in ProTraits is provided with a confidence score between 0 and 100%, whereas 100% denotes the highest confidence for the respective phenotype of a specific organisms. Here, we used only the carbon source utilisation phenotypes with the stringent confidence threshold of $\geq$ 95%. The data was directly downloaded from the ProTraits website (http://protraits.irb.hr/data.html) as a tab-separated table. For organisms which had at least one high-confidence carbon source utilisation trait, the corresponding genome assembly was obtained from NCBI RefSeq [105] if available. In cases where a genome assembly was found, it was applied as input for ModelSEED, CarveMe, and `gapseq` to reconstruct metabolic models. The number of potential carbon sources was reduced to a subset for which a mapping from substance name to ModelSEED and CarveMe model namespace existed (`dat/sub2pwy.csv`).

The tests for D-lyxose were removed because it was listed as all negative in ProTraits and also all compared pipelines predicted no utilisation. The main test whether a carbon source can be used by a model was done in a BIOLOG-like manner as described above (see Gap-filling of biomass, carbon sources, and fermentation products). To this end, temporary reactions to recycle reduced electron carriers as carbon source utilisation indicators were added to the respective model. The objective for optimisation was set to maximise the flux through these recycling reactions. The exchange reactions were limited to a minimal medium with minerals and the focal potential carbon source. This theoretical approach tested, whether the model is able to pass electrons from the potential carbon source to electron carrier metabolites. A carbon source was predicted to be able to serve as energy source if at least one of the recycle reactions carried a positive flux.

### Prediction of gene essentiality

Genome-scale metabolic models are commonly used to predict if essentiality of genes for cellular growth. In order to further evaluate our `gapseq` approach, we compared gene essentiality predictions with previously reported growth phenotypes of single gene-knockout experiments. To predict the essentiality of genes, we performed in silico single gene deletion phenotype analysis for the network reconstructions of *Escherichia coli* str. K-12 substr. MG1655 (RefSeq assembly accession: GCF_000005845.2), *Bacillus subtilis* substr. *subtilis* str. 168 (GCF_000789275.1), *Shewanella oneidensis* MR-1 (GCF_000146165.2), *Pseudomonas aeruginosa* PAO1 (GCF_000006765.1), and *Mycoplasma genitalium* G37 (GCF_000027325.1). The analysis was performed on the basis of the models' Gene-Protein-Reaction (GPR) mappings and according to the protocol by Thiele and Palsson, 2010 [22]. To this end, the contingency tables of predicted growth/no-growth phenotypes from the network models and experimentally determined growth phenotypes of gene deletion mutants were constructed. Genes were predicted to be conditionally essential under the given growth environment if the predicted growth rates of the models were below $0.01 \text{ h}^{-1}$. The growth-media compositions for growth predictions were defined as M9 with glucose as carbon- and energy source for *E. coli*, lysogeny broth (LB) for *B. subtilis* and *S. oneidensis*, M9 with succinate as carbon and energy source for *P. aeruginosa*, and a complete medium (all external metabolites available for uptake) for *M. genitalium*. Experimental data for gene essentiality was obtained from [106–110]. In order to compare GPR associations between reconstructions, it was tested if the GPR boolean expressions from two models for the same enzymatic reaction return identical results for all possible combinations of gene presence/knockout of the involved genes.

### Fermentation product tests

To evaluate the potential of our approach to predict bacterial metabolism in anaerobic environments, we simulated the anoxic growth of bacterial model reconstructions and compared the predictions with fermentation end products reported in primary literature. The release of by-products from anaerobic metabolism was predicted using Flux Balance Analysis (FBA) coupled with a minimisation of total flux [111] to avoid fluxes that do not contribute to the objective function of the biomass production. In addition, Flux-Variability-Analysis (FVA) [112] was applied to predict the maximum fermentation product release of individual metabolites across all possible FBA solutions.

Metabolites with a positive exchange flux (i.e. outflow) were considered as fermentation products. The analysis was performed for 24 different bacterial organisms, which (1) have a genome assembly available in the RefSeq database [105], (2) are known to grow in anaerobic environments, and (3) for which the fermentation products have been described in the literature based on anaerobic cultivation experiments (Additional file 2: Table S2). The gap-filling of the network models using gapseq, CarveMe, and ModelSEED as well as the simulations of anaerobic growth were all performed assuming the same growth medium that comprised several organic compounds (i.e. carbohydrates, polyols, nucleotides, amino acids, organic acids) as potential energy sources and nutrients for growth (see media file dat/media/FT.csv at the gapseq github repository).

Since the amount of fermentation product release depends on the organism's growth rate, we normalised the outflow of the individual fermentation products, which has the unit $mmol * gDW^{-1} * h^{-1}$, by the predicted growth rate of the respective organism which has the unit $h^{-1}$. Thus, we report the amount of fermentation product production in the quantity of the metabolite that is produced per unit of biomass: $mmol * gDW^{-1}$.

### Pathway prediction of soil and gut microorganisms

The pathway analysis was done by comparing predicted pathways of soil and gut microorganisms. For this means, genomes were downloaded from a resource of reference soil organisms [113] and gut microorganisms [68]. The default parameter of gapseq were used for pathway prediction. The principal component analysis was done in R using the factoextra package [114]. For predicted pathways for soil and gut microorganisms, it was checked if samples belong to different distributions using a bootstrap version of the Kolmogorov-Smirnov test [115].

### Anaerobic food web of the human gut microbiome

Microbial strains rarely live in isolation but usually coexists with other strains in multi-species communities. In such communities, metabolic processes in one organism may affect the metabolism of other cells and vice versa [116]. It is an ambitious goal in systems biology to apply genome-scale metabolic models in simulations of community metabolism, including metabolite exchanges between cells of different species. Here, we evaluated the potential of automatically reconstructed models to predict metabolic interactions in an anaerobic microbial community. As a test case, representative bacterial organisms known to be relevant in the human intestinal cross-feeding of metabolites were selected based on the proposed food webs by Louiset al. (2014) [50] and Rivera-Chavez et al. (2015) [51]. The genomes of organisms were obtained from NCBI RefSeq [105] and metabolic models reconstructed using gapseq, carveme, and modelseed. A medium containing minerals, vitamins, amino acids, fermentation- and metabolic by-products (namely acetate, formate, lactate, butyrate, propionate, $H_2$, $CH_4$, ethanol, $H_2S$, succinate), and carbohydrates (glucose, fructose, arabinose, ribose, fucose, rhamnose, lactose) was used for gap-filling. Furthermore, a published model of *Methanosarcina barkeri* was added to the community [117] to represent archaea that are also known to be part of anaerobic food webs [118]. All organisms of the modelled community and their respective genome assembly accession numbers are listed in Additional file 2: Table S3. In detail, all metabolic models were simulated as microbial community using the R-package 'BacArena', which allows an individual-based dynamic simulation of metabolic

models that are optimised separately in a shared growth environment [52]. A virtual growth environment ('arena') of the size of 20 × 20 grid cells was defined. For each organism of the microbial community, five random grid cells within the arena were populated with the model of the focal strain to define the initial community composition. The gap-filling medium described above, but without the fermentation and by-products was used to determine the initial arena substance concentrations. In addition, sulfite and 4-aminobenzoate were added in 1 mM each to the growth environment as these metabolites are essential for the growth of the *M. barkeri* model. Subsequently, the community was simulated for seven time steps, which corresponds to seven hours simulation time. The metabolite uptake and production rates were analysed after the third time step for CarveMe and `gapseq` models and after the fifth time step for ModelSEED models, in which all organisms were growing exponentially and reached similar total population density.

### Model reconstructions from metagenomic assemblies

Genomes assembled from metagenomic data via 'binning' are often fragmented and incomplete. For the reconstruction of metabolic models from such genomes, it is important to estimate how missing genetic fragments may affect final model quality. 4930 species-level genome bins (SGBs) assembled from shotgun metagenome sequencing reads were obtained from the study of Pasolli et al. (2019) [61]. Only those SGBs were considered for further analysis, which were already classified as bacteria on a species-level in the original publication. For each SGB, closely related reference assemblies from the RefSeq database [105] were identified by constructing a multi-locus phylogenetic tree using autoMLST (version as of April 7, 2020, [119]). RefSeq assemblies were considered as genomes from the same species-level taxonomic group as the focal SGB if their predicted MASH distance ($D$) [120] were below or equal to 0.05. This threshold was shown before to cluster bacterial genomes at the taxonomic level of species [120]. Only SGBs with 10 or more assigned reference assemblies were considered for further analysis, which yielded in total 127 SGBs. Metabolic models were reconstructed using `gapseq` for each SGB and their 10 closest reference assemblies (Additional file 2: Table S5). Next, similarity of SGB models with their respective reference models was calculated using the following metabolic network similarity score $T_{\text{SGB}}$:

$$T_{\text{SGB}} = \frac{\sum_i a_i b_i}{\sum_i b_i} \quad , \quad i \in R_{\text{SGB\_Ref}} \ , \ 0 \ \leq b_i \leq 1$$

with

$$a_i = \begin{cases} 0 & \text{if } i \ \notin R_{\text{SGB}} \\ 1 & \text{if } i \ \in R_{\text{SGB}} \end{cases}$$

(2)

$R_{\text{SGB\_Ref}}$ is the union set of reactions with associated genes that are part of the network models reconstructed for the ten reference genome assemblies of the focal SGB. $R_{\text{SGB}}$ is the set of reactions part of the SGB's model reconstruction. $b_i$ is the frequency of reaction $i$ among the ten SGB's reference models. Completion of the genome sequence of SGBs was estimated by using BUSCO (version 4.0.6, [104]) using the lineage-specific completeness score.

### Technical details

The pathway prediction part of `gapseq` is implemented as Bash shell script and the metabolic model generation part is written in R. Linear optimisation can be performed with a different solvers (GLPK or CPLEX). Other requirements are exonerate, bedtools, and barrnap. In addition, the following R packages are needed: data.table [121], stringr [122], sybil [65], getopt [123], reshape2 [124], doParallel [125], foreach [126], R.utils [127], stringi [128], glpkAPI [129], and BioStrings [130]. Models can be exported as SBML [131] file using sybilSBML [65] or R data format (RDS) for further analysis in R, for example with sybil [65] or BacArena [52].

## Supplementary Information

---

**Additional file 1:** Supplementary figures S1-S5.

**Additional file 2:** Supplementary tables S1-S7. OpenDocument spreadsheet (ODS) file

**Additional file 3:** Review history.

---

#### Acknowledgements
We thank Martin Sperfeld for fruitful comments and discussions during the developmental phase. The software was thankfully tested by Georgios Marinos, Shan Zhang, and Lena Best.

#### Review history
The review history is available as Additional file 3.

#### Peer review information

#### Authors' contributions
All authors conceptualised `gapseq` and wrote the manuscript. JZ and SW developed the software and did the analysis. The authors read and approved the final manuscript.

#### Authors' information
Twitter handles: @_jozimmermann (Johannes Zimmermann); @KaletaLab (Christoph Kaleta); @SWaschina (Silvio Waschina).

#### Funding
CK and SW acknowledges support by the Collaborative Research Centre 1182 - 'Origin and Function of Metaorganisms' - Deutsche Forschungsgemeinschaft and by the Cluster of Excellence 2167 - 'Precision medicine in chronic inflammation' - Deutsche Forschungsgemeinschaft. In addition, CK acknowledges support by the German Ministry for Education and Research within the context of iTREAT (BMBF support code 01ZX1902A). The funders had no role in study design, data collection and analysis, decision to publish, or preparation of the manuscript. Open Access funding enabled and organized by Projekt DEAL.

#### Availability of data and materials
`gapseq` is implemented as bash shell script and in R and is freely available under the GNU General Public License (v3.0) on GitHub (https://github.com/jotech/gapseq/). Documentation and tutorials for gapseq can be found at https://gapseq.readthedocs.io. All results presented in this manuscript were produced using the specific `gapseq` version 1.1 as archived on GitHub [63] and is available from Zenodo [132]. The datasets used for model construction and validation purposes were obtained from publicly available databases and publications as cited at the respective parts of the manuscript. Scripts and data used for the benchmarking tests in this study are available from the GitHub repository, https://github.com/Waschina/gapseqEval.

#### Ethics approval and consent to participate
Not applicable.

#### Consent for publication
Not applicable.

#### Competing interests
The authors declare that they have no competing interests.

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

## 