## [**Additional file 3** Review history. · Genome Biology]

Review History

First round of review

Reviewer 1

Were you able to assess all statistics in the manuscript, including the appropriateness of statistical tests used? No, I do not feel adequately qualified to assess the statistics.

Were you able to directly test the methods? No.

Comments to author:

-Is the Software likely to be of broad utility?

Yes

Is it easy to install and use?

At least for Windows users, it will not be straightforward. I didn't test the software.

-Does the software represent a significant advance over previously published tools (as demonstrated by direct comparison with available related software)?

The authors didn't compare their pipeline with important reconstruction tools available. Suggestions and more details can be found below

-Please indicate briefly the novel features and/or advantages of the software and/or please reference the relevant publications and which alternatives, if any, it should be compared with.

The pipeline presented by the authors show an increased accuracy of predictions, especially for the formation of by-products. This is quite relevant, in especial to model microbial communities.

Also, I think it is quite interesting the approach. Pathway prediction, transport prediction and protein complex prediction are not new methods. These were implemented by Pathway Tools decades ago (not exactly the same but the ideas is the same). However, they merge these algorithms with an excellent database (ModelSEED), and that novel mix (excellent algorithms + excellent database) results in an increased predictive power. This is a certainly an advance in the field.

-Are the methods appropriate to the aims of the study, are they well described, and are necessary controls included?

They are appropriate. Below, I describe which sections needs to be improved

-Are the conclusions adequately supported by the data shown?

Yes

-Are sufficient details provided to allow replication and comparison with related analyses that may have been performed?

Suggestions to provide more details in the section of methods can be found below. I would suggest that the authors verify that they uploaded the performed in silico tests to the github repository. In particular, it is important that they show the files with the experimental data used to validate their pipeline

-Is the paper of broad interest to others in the field, or of outstanding interest to a broad audience of

biologists?

It has a strong bioinformatics background. I suggest some changes to make it suitable for a broader audience.

The authors presented a new pipeline for pathway prediction and genome-scale metabolic reconstruction. This new pipeline certainly advances the field of metabolic reconstruction and therefore this article is suitable for publication in genome biology. However, several major concerns arise and I would invite the authors to carefully read and take care of all the comments, questions and suggestions. To solve these issues will lead to the creation of an article which is clear to a broad audience of readers and strong in its foundation.

The first thing that unpleasantly surprises me is that only two tools in the literature are used for the benchmarking. Even more, 5 tools for genome-scale metabolic reconstruction are not even cited in the manuscript (RAVEN, Merlin, AuReMe, AutoKEGGRec, MetaDraft). This is a major fault. You should at least mention in the introduction the existence of other reconstruction tools to inform the readers of current alternatives to your pipeline and why there is a need to create a new tool.

I understand why you are eager to use only two tools for benchmarking. ModelSEED and Carveme are the only tools that generate models with positive flux through the biomass equation. Thus, the models are ready to predict phenotypes. However, when you omit other tools from the comparison, you implicitly say that other tools cannot be compared. Recently, a paper, which was also published in the journal Genome Biology (<https://genomebiology.biomedcentral.com/articles/10.1186/s13059-019-1769-1>), presented several criteriums to independently assess a reconstruction tool, even when the tools are not able to generate biomass. You should use the criteriums (or at least the most relevant) presented in this paper to make a comparison table to compare your pipeline with other tools in order to inform readers about the advantages/disadvantages of your tool.

Finally, I noted that you used the ModelSEED database in your pipeline (retrieved in January 2018).

There is a preprint presenting the ModelSEED database as a resource that can be used for research (<https://www.biorxiv.org/content/10.1101/2020.03.31.018663v2.full.pdf>). Even though you retrieved an older version of the database, you should cite this preprint in order to give credit to the creators of the database (which are not exactly equal to the creators of reconstruction tool)

Please, cite the literature thoroughly.

Lines 435-436. It is important to distinguish between metabolic pathways and subsystems. A subsystem "is a set of functional roles that together implement a specific biological process or structural complex. A subsystem may be thought of as generalization of the term pathway" (quote from reference 1). Metacyc and KEGG work with pathways and ModelSEED with subsystems.

First, it is confusing that the authors use the term pathway database for ModelSEED because ModelSEED does not work with pathways in the same strict way than Metacyc or KEGG do. I see that you define in line 435 the concept of pathway as a list of reactions with enzyme names and EC numbers. However, that definition is very ambiguous because subsystems from ModelSEED are wider than pathways from Metacyc. Furthermore, because of the high heterogeneity of pathway definitions between databases, it is not clear to me how the authors decided which definition was correct? Depending on the pathway definition, results may vary during the pathway prediction procedure (lines 452-475). Did the authors use the pathway prediction for each database in sequential order (for example, first Metacyc, then KEGG, then MODELSEED) or they manually selected pathways for this process in to avoid different pathways definition? If the process is performed in sequential order, do the results depend on the order or the databases selected? If the process is sequential, which database order was used? The procedure is not very clear and I would appreciate that you provide more details of the process in point 4.3

445: are these amino acid sequences or nucleotide sequences?

Lines 445-451. When you do this, you have to map reactions to sequences using intermediate information (EC numbers, enzyme names, gene names). This process of mapping could be very chaotic as:

1) names could vary between databases and the string comparison will fail because of small differences

in names.

2) Reactions are not always associated with EC numbers or other data. Even when they are associated to EC numbers, EC numbers could be not specific enough.

You should describe how you dealt with these cases and also to describe your database a bit deeper. For example, how many sequences are associated to each reaction? How many reactions could be mapped through EC numbers?

In addition, is it common to find a reaction that could be associated to multiple EC numbers. In that case, you would associate sequences from each EC number to that reaction. Could this big amount of different types of sequences result in the incorporation of false-positive reactions to the network?

Lines 464-472: From the total of evaluated genomes how many times per genome a pathway is predicted to be completed because of criteriums 1 (lines 466-469) or 2 (469-472) (in average)?

Lines 479-480: what happens with FASTA files that do not have that information? Some tools such as ModelSEED are able to predict protein complexes and do not need headers. They only need protein sequences. I think that the approach presented by the authors limits the application of the pipeline because there are different genome annotation pipelines and the headers in the output files of these pipelines are not homogenous among them. Therefore, you should specify the compatibility with annotation pipelines. I guess that you used the structure used in the genome annotations from NCBI.

Lines: 484-486: "could not be matched" instead of "could not matched". In general, the paper is well written. There are a few other mistakes. Please check the manuscript thoroughly.

Lines 484-486: different verb tenses are used. "Were" should be used instead of "are" in line 485

Lines 493-521: In many cases, TC Database has general transporters. If you reduce the TC database considering only metabolites that match those known to be produced or consumed by microorganisms, you will remove these general transporters which may also be useful. How many did you remove?

479-481: I think this should be evaluated. The BiGG database is a collection of models that has been manually curated. Each model has reactions with gene-protein-reaction (GPR) associations. In the BiGG database some reactions are catalyzed by protein complexes, which is represented as a GPR with the logical operator & (and). For example, the GPR for reaction GLYCTO2 (<http://bigg.ucsd.edu/models/iML1515/reactions/GLYCTO2>) is "b2979 and b4468 and b4467". What is the degree of agreement between GPRs of BiGG and your database?

Line 500: determining the substrate type using the header is OK. However, as mentioned for lines 479-480, this limits the application of the pipeline.

Lines 528-538: A big effort has been made by the ModelSEED team during the last years to improve and curate their database [reference 2]. We see that the ModelSEED database was retrieved more than 2 years ago (line 525). The inconsistencies mentioned by the authors and many others were corrected in the new ModelSEED database. Therefore, this is a duplication of efforts. Can the new ModelSEED database be easily incorporated into the pipeline?

Lines 555-574: The approach of obtaining EC numbers, TC numbers and annotations from the genome limits the application of the pipeline as mentioned before.

The processes of pathway prediction and gap-filling seems redundant in the sense that one could avoid the process of pathway prediction and just run the gap-filling procedure after the homology search in order to create a network able to generate biomass (like in Carveme). One could just add to the draft model reactions that have a homology match and then run the gap-filling algorithm. The pathway prediction process is not perfect and therefore some reactions that shouldn't be added will be added to the draft model. Why not skip this process at all and just run a gap-filling algorithm where the minimum amount of reactions needed to make biomass are added to the network? One of the main problems with ModelSEED (at least the old version) and CarveMe is that they return networks with too many reactions which shouldn't be in the reconstruction. It seems that the same problem is in the author's pipeline and therefore, their method is not solving the main problem of current reconstruction tools. I would appreciate your comments. If necessary, try to explain this somewhere in the manuscript.

575- 610: the gap-filling algorithm is quite clever.

618-619: Here, it says that the gap-filling procedure is performed in 4 steps. I don't understand why you

need four steps. the goal of the gap-filling procedure is just one: to achieve biomass production. Steps 2 to 4 are not necessary.

629-631. With this approach, you could add more reactions than the ones actually needed. Again, wouldn't the networks created with this pipeline have too many reactions? On average, how big are the genome-scale metabolic reconstructions built with this pipeline?

Do users have to define which alternative energy sources can be used by the microorganism? If the user does not provide data, then how the pipeline decide which carbon sources should sustain in silico growth? This should only be performed if the user provides data. Otherwise, it would lead to false-positive results. Why M9 media is used? Shouldn't you use a user-defined media? Why this is performed only for carbon sources and not for nitrogen sources?

Line: 637: what do you mean with artificial? That's not clear. Please, specify.

Line 642: what do you mean with "the same list of compounds as for step (3)"? Are you referring to ubiquinol, menaquinol, or NADH? In general, this part of gap-filling is not very clear. Please, provide more details.

Lines 678-680: the sentence is strangely written. Maybe you could re-phrase it.

678-688: Could this approach be biased? I wonder how complete are the annotations (in terms of EC numbers) for the model files created with Carveme and ModelSEED. The authors map reactions through EC numbers. What happens if a model file created with Carveme does not have many annotations in terms of EC numbers. Then, the match between the EC numbers in the genome and the metabolic model would be very poor and the conclusion obtained with the author's pipeline would be that the model generated with Carveme is very poor. However, the reality is that reactions are there, but the annotations (EC numbers) are missing. Does the pipeline of the authors does consider this type of bias? Please comment.

685-688: I don't understand what kind of data is present in BacDive. In line 664, you only mentioned that you used enzymatic data, but readers would appreciate that you could specify which kind of data can be found in that database. What "active" and "not active" mean? Is this measured enzymatic activity experimentally collected for strains grown on a particular media condition?

Line 699-701: I don't understand why you removed D-lyxose. If all the pipelines predict no utilization and that agree with experimental data, it is OK to include it.

Lines 702-704: This seems very important to understand the performed in silico test. It is very interesting the approach that you used because it is indeed closer to the experimental procedure of BIOLOG arrays than when you maximize growth rate. Can you explicitly describe the reaction equations of the reactions that you temporarily added to the model?

Line 722: how changes in this value affect the results?

728-750: this looks OK.

772: Is this a dynamic simulation?

how the authors modeled the microbial communities? I think a method section describing the details for the community modeling is missing.

I would suggest including an introductory paragraph in each section of methods. These paragraphs should mention the general idea of the section.

RESULTS

Lines 85-86: Why you removed dead-end metabolites and the corresponding reactions. Even though reactions that have unconnected metabolites do not carry flux, they could be important to complete models as knowledge-bases.

Lines 94-95: these 10,538 enzyme activities correspond to what? Do these 10,538 enzyme activities come from different microorganisms? Please specify in the method section (and it would be also useful in the results section) for which organisms and which conditions these enzymatic activities were collected. How many EC numbers are related to those 10,538 enzymatic activities?

Line 100: How do you define false negative? I guess that EC number present in the genome annotation but not present in the metabolic model?. You should explicitly define this in the method section.

Line 102: this is very strange. 26% of the comparisons means $10,538 * 26 / 100 = 2739$ comparisons. That means that almost all the microorganisms had the enzymatic activity 1.11.1.6. Is that correct? I wonder about the distribution of EC numbers counts (sorted frequency of EC numbers). It seems that it could be biased for certain EC numbers. How this bias could affect the results?

Lines 113-120: you describe in the methods section that you used data from ProTraits. However, you didn't describe what type of data is in this database. Do the data from ProTraits come from BIOLOG arrays? Was this data collected assessing growth on flasks by measuring OD? Was this data collected using both techniques? You implemented an in silico test which is analogous to the BIOLOG test. You can only compare results from this in silico test with data that was collected with BIOLOG arrays. One can infer from the explanation that you gave for formate overestimation in lines 113-120 that the experimental data for formate was a measure of formate being incorporated into the biomass components (for example one could test this in a flask with formate as sole carbon source and by measuring OD). If that's the case, you should perform an in silico test where you actually maximize growth rate and not on the basis of electron transfer.

Results from sections 2.4 and 2.5 are very impressive. It is quite shocking how bad carveme and modelseed performs in terms of by-product prediction.

Lines 174-176. You should briefly describe the community modeling framework BacArena. Some basic questions that you should answer are

- 1) Is this a dynamic simulation or is a steady-state simulation? It seems that it is dynamic because you mention time step in line 175 but you should make this explicit.
- 2) What was the objective function of the microbial community? Or each model was optimized separately in a common environment?
- 3) What are the inputs and outputs of this framework?

Looking at methods (lines 775-777), why did you choose the third time step? It would be good that you actually show the predicted uptake and production for each time step until the final hour. Why you show uptake and production rates and not changes in metabolite concentrations? This would make more sense as it is a dynamic simulation.

References

- 1) Overbeek, R., Begley, T., Butler, R. M., Choudhuri, J. V., Chuang, H. Y., Cohoon, M., de Crécy-Lagard, V., Diaz, N., Disz, T., Edwards, R., Fonstein, M., Frank, E. D., Gerdes, S., Glass, E. M., Goessmann, A., Hanson, A., Iwata-Reuyl, D., Jensen, R., Jamshidi, N., Krause, L., ... Vonstein, V. (2005). The subsystems approach to genome annotation and its use in the project to annotate 1000 genomes. *Nucleic acids research*, 33(17), 5691-5702. <https://doi.org/10.1093/nar/gki866>

2) Samuel M. D. Seaver, Filipe Liu, Qizhi Zhang, James Jeffryes, José P. Faria, Janaka N. Edirisinghe, Michael Mundy, Nicholas Chia, Elad Noor, Moritz E. Beber, Aaron A. Best, Matthew DeJongh, Jeffrey A. Kimbrel, Patrik D'haeseleer, Erik Pearson, Shane Canon, Elisha M. Wood-Charlson, Robert W. Cottingham, Adam P. Arkin, Christopher S. Henry

bioRxiv 2020.03.31.018663; doi: <https://doi.org/10.1101/2020.03.31.018663>

===

Reviewer 2

Were you able to assess all statistics in the manuscript, including the appropriateness of statistical tests used? No, I do not feel adequately qualified to assess the statistics.

Were you able to directly test the methods? Yes.

Comments to author:

The manuscript entitled "gapseq: Informed prediction of bacterial metabolic pathways and reconstruction of accurate metabolic models" by Johannes Zimmermann, Christoph Kaleta and Silvio Waschina, is a computational paper focused on the development of a software that assists in the automated reconstruction of genome-scale metabolic models, integrating a vast amount of information collected through databases.

This is a solid work, useful for the community, that presents a new approach to overcome the major pitfalls of the current genome-scale metabolic reconstruction approaches. Many details are reported, to inform the reader about how the pipeline of the work has been envisioned, which data have been retrieved, and how these were used. The manuscript is very well written, with concepts being clearly presented.

Through this elegant effort, metabolic models that are reconstructed using gapseq may be used to generate models where multiple species co-exist, such the environment observed in the microbiome.

Below I highlight a couple of major point for improvement as well as some minor comments, which should be addressed before publication:

Major comments:

- Page 3, line 78. The author state that "In addition, the protein sequence database in gapseq can be updated to include new sequences from Uniprot and TCDB." Is the update automatic or manual? To make the software useful for the community, the update should be automatic, so that the most recent updates may be available when the user launches a job through the software.

- Page 7, line 176. The authors state that "On the community level, simulations using gapseq models captured all important substances, which are known to be produced in the context of the food web (Figure 4)". While gapseq largely outperforms regarding the production of important substances, the consumption appears to be not recapitulated by gapseq. However, the authors do not highlight this and, at Page 7, line 213 they state that "The overall consumption pattern and individual microbial contributions were found to be in agreement with literature data.". This statement is not supported by the outcome of Figure 4, and the authors should develop further on the outcome presented in Figure 4 regarding the consumption across CarveMe, gapseq and ModelSEED, and identify the reasons of the decrease of gapseq in capturing the consumption.

Minor comments:

- Page 2, line 16. The authors state that "Metabolic models not only have demonstrated their ability to predict phenotypes on the level of cellular growth and gene knockouts, but also provide potential molecular mechanisms in form of gene and reaction activities, which can be validated experimentally [5]". The reference is outdated; following the reasoning of the authors, it would be appropriate to indicate more recent modelling efforts which have led to an experimental validation of some among those molecular mechanisms.
- Page 3, line 66. The authors state that "Topology as well as sequence homology to reference proteins inform the filling of network gaps, and the screening for potential carbon sources and metabolic products is done in a way that reduces the impact of growth medium definitions." It would be better to indicate, shortly, how this is done, rather than state "...is done in a way that...".
- Page 9, line 258. The authors state that "For each validation approach, predictions were compared to experimental data obtained from databases and literature to calculate prediction performance scores.". It would be useful to mention which type of data from the literature and which databases have been used for the validation, so that the reader will be immediately aware - here in this paragraph - of where to retrieve the relevant information.
- For some reason, the references do not appear in the text in a progressive numbering. This should be fixed.
- Some of the articles listed in the references include information that do not pertain to the article(s), such for example the manuscripts cited that have appeared on Current Opinion in Systems Biology. The correctness of the information should be double-checked.
- Figure S2 is indicated in the main text, but Figure S1 is not.
- Typos:
 - Page 5, line 126. Please, correct "essentially" with "essentiality" .
 - Page 5, line 135. Please, remove the comma after "...to the fact".
 - Page 5, line 145. "Figure 2" should be "Figure 3".
 - Page 13, line 369. Please, remove the comma after "Although".

Reviewer #1

[Comment - 1]

Is the Software likely to be of broad utility?

Yes

Is it easy to install and use?

At least for Windows users, it will not be straightforward. I didn't test the software.

Does the software represent a significant advance over previously published tools (as demonstrated by direct comparison with available related software)?

The authors didn't compare their pipeline with important reconstruction tools available. Suggestions and more details can be found below

Please indicate briefly the novel features and/or advantages of the software and/or please reference the relevant publications and which alternatives, if any, it should be compared with.

The pipeline presented by the authors show an increased accuracy of predictions, especially for the formation of by-products. This is quite relevant, in especial to model microbial communities.

Also, I think it is quite interesting the approach. Pathway prediction, transport prediction and protein complex prediction are not new methods. These were implemented by Pathway Tools decades ago (not exactly the same but the ideas is the same). However, they merge these algorithms with an excellent database (ModelSEED), and that novel mix (excellent algorithms + excellent database) results in an increased predictive power. This is a certainly an advance in the field.

Are the methods appropriate to the aims of the study, are they well described, and are necessary controls included?

They are appropriate. Below, I describe which sections needs to be improved

Are the conclusions adequately supported by the data shown?

Yes

Are sufficient details provided to allow replication and comparison with related analyses that may have been performed?

Suggestions to provide more details in the section of methods can be found below. I would suggest that the authors verify that they uploaded the performed in silico tests to the github repository. In particular, it is important that they show the files with the experimental data used to validate their pipeline

[Answer]

Data files and scripts to perform the evaluation test are now available at the GitHub repository <https://github.com/Waschina/gapseqEval>

[Comment - 2]

Is the paper of broad interest to others in the field, or of outstanding interest to a broad audience of biologists?

It has a strong bioinformatics background. I suggest some changes to make it suitable for a broader audience.

[Answer]

We thank the reviewer for this advice. We reworded large parts of the manuscript in the course of the revisions as you can see in the manuscript version with marked changes. In doing so and where possible, we aimed to explain our approach and concepts to a broader readership that may not be familiar with the terminology in bioinformatics.

[Comment - 3]

Comments

The authors presented a new pipeline for pathway prediction and genome-scale metabolic reconstruction. This new pipeline certainly advances the field of metabolic reconstruction and therefore this article is suitable for publication in genome biology. However, several major concerns arise and I would invite the authors to carefully read and take care of all the comments, questions and suggestions. To solve these issues will lead to the creation of an article which is clear to a broad audience of readers and strong in its foundation.

[Answer]

We thank the reviewer for the comprehensive and careful assessment of our manuscript. Please find our answers to each comment below. Indicated line numbers refer to the position in the revised manuscript with tracked changes.

[Comment - 4]

The first thing that unpleasantly surprises me is that only two tools in the literature are used for the benchmarking. Even more, 5 tools for genome-scale metabolic reconstruction are not even cited in the manuscript (RAVEN, Merlin, AuReMe, AutoKEGGRec, MetaDraft). This is a major fault. You should at least mention in the introduction the existence of other reconstruction tools to inform the readers of current alternatives to your pipeline and why there is a need to create a new tool.

[Answer]

We agree with the reviewer, that mentioning other reconstructions tools should be an essential part of the introduction. In the revised manuscript, a new paragraph was integrated, where we elaborate on other available tools (lines 33-50). In the introduction, we now highlight also the potential to predict metabolic physiology of microorganisms from automated network reconstructions and the challenges that currently still limit the broad application of reconstruction tools, especially in the context of microbial community metabolism (lines 51-60). Additionally, we now also highlight already in the introduction why

(direct reconstruction of models with positive biomass reaction flux) the extensive benchmarking was done against the two tools CarveMe and ModelSEED (lines 71-78 and 114-117).

[Comment - 5]

I understand why you are eager to use only two tools for benchmarking. ModelSEED and Carveme are the only tools that generate models with positive flux through the biomass equation. Thus, the models are ready to predict phenotypes. However, when you omit other tools from the comparison, you implicitly say that other tools cannot be compared. Recently, a paper, which was also published in the journal Genome Biology (<https://genomebiology.biomedcentral.com/articles/10.1186/s13059-019-1769-1>), presented several criteriums to independently assess a reconstruction tool, even when the tools are not able to generate biomass. You should use the criteriums (or at least the most relevant) presented in this paper to make a comparison table to compare your pipeline with other tools in order to inform readers about the advantages/disadvantages of your tool.

[Answer]

We thank the reviewer for pointing out the study by Mendoza *et al.* (2019) in Genome Biology. As suggested, a new subsection under Results has been added that compares the reconstruction tools that were also part of the study by Mendoza *et al.* with our *gapseq* approach. Therefore, we focused on the criteria that will be most relevant for potential users to decide whether *gapseq* is suitable for their specific research question, or if another tool would be more fitting. The new subsection has the numbering 2.10. and the comparison is summarised in Table 2.

[Comment - 6]

Finally, I noted that you used the ModelSEED database in your pipeline (retrieved in January 2018). There is a preprint presenting the ModelSEED database as a resource that can be used for research (<https://www.biorxiv.org/content/10.1101/2020.03.31.018663v2.full.pdf>). Even though you retrieved an older version of the database, you should cite this preprint in order to give credit to the creators of the database (which are not exactly equal to the creators of reconstruction tool)

[Answer]

We thank the reviewer for pointing out the ModelSEED database preprint and we are pleased to see that in the meantime the manuscript appeared as publication in Nucleic Acid Research (<https://doi.org/10.1093/nar/gkaa746>). We referenced the publication by Seaver *et al.* in the revised version of our manuscript.

[Comment - 7]

Please, cite the literature thoroughly.

[Answer]

A number of additional studies, including all publications mentioned by both reviewers, are now cited.

[Comment - 8]

Lines 435-436. It is important to distinguish between metabolic pathways and subsystems. A subsystem "is a set of functional roles that together implement a specific biological process or structural complex. A subsystem may be thought of as generalization of the term pathway" (quote from reference 1). Metacyc and KEGG work with pathways and ModelSEED with subsystems.

First, it is confusing that the authors use the term pathway database for ModelSEED because ModelSEED does not work with pathways in the same strict way than Metacyc or KEGG do. I see that you define in line 435 the concept of pathway as a list of reactions with enzyme names and EC numbers. However, that definition is very ambiguous because subsystems from ModelSEED are wider than pathways from Metacyc. Furthermore, because of the high heterogeneity of pathway definitions between databases, it is not clear to me how the authors decided which definition was correct? Depending on the pathway definition, results may vary during the pathway prediction procedure (lines 452-475). Did the authors use the pathway prediction for each database in sequential order (for example, first Metacyc, then KEGG, then MODELSEED) or they manually selected pathways for this process in to avoid different pathways definition? If the process is performed in sequential order, do the results depend on the order or the databases selected? If the process is sequential, which database order was used? The procedure is not very clear and I would appreciate that you provide more details of the process in point 4.3

[Answer]

We thank the reviewer for pointing out the importance of the definition of pathways. We agree that the definition has a strong impact on which reactions and metabolic routes are predicted. By default, *gapseq* uses the pathway topologies and 'base pathway' definition from MetaCyc. This is because MetaCyc has, in our opinion, the strictest definition of pathways:

1. Experimentally determined metabolic routes;
2. small molecule (metabolite) pathways;
3. curated from scientific literature [Caspi et al. (2012) *Nucl. Acids R.*]; and
4. the *base pathways* definition: "pathways composed of reactions only, where no portion of the pathway is designated as a subpathway" [Caspi et al. (2019) *Nucl. Acids. R.*].

Especially (4) makes the pathway predictions in *gapseq*, which incorporates also the pathway's completion, most suited for the MetaCyc Pathway definition. In addition, the advantage of using MetaCyc's pathways is that they often include information on the metabolic routes' key enzymes.

While we highly recommend to use *gapseq*'s default for MetaCyc pathway topologies, we decided to keep the option for users to alternatively use the pathways from KEGG or subsystem definitions from ModelSEED. This could for instance be of interest for users, who intend to use solely *gapseq*'s first step (i.e. the prediction of enzymes and pathway/subsystem coverage) for pathway analysis covering reference pathways/subsystems other than MetaCyc.

We updated sections 4.2 and 4.3 and elaborated on which pathway definition *gapseq* uses in default settings, which alternative options exist, and why the default setting is recommended.

[Comment - 9]

445: are these amino acid sequences or nucleotide sequences?

[Answer]

These are amino acid sequences for the reference proteins. We clarified the type of those sequences (subsection 4.2.).

[Comment - 10]

Lines 445-451. When you do this, you have to map reactions to sequences using intermediate information (EC numbers, enzyme names, gene names). This process of mapping could be very chaotic as:

- 1) names could vary between databases and the string comparison will fail because of small differences in names.
- 2) Reactions are not always associated with EC numbers or other data. Even when they are associated to EC numbers, EC numbers could be not specific enough.

You should describe how you dealt with these cases and also to describe your database a bit deeper. For example, how many sequences are associated to each reaction? How many reactions could be mapped through EC numbers?

In addition, is it common to find a reaction that could be associated to multiple EC numbers. In that case, you would associate sequences from each EC number to that reaction. Could this big amount of different types of sequences result in the incorporation of false-positive reactions to the network?

[Answer]

In order to avoid a chaotic mapping we employed the following steps: i) EC numbers are preferred when possible, ii) MetaCyc offers curated sequence association (Uniprot IDs) for many reactions, iii) if no EC number or curated association was found, then the Enzyme commission reaction names were used.

In detail, we focused on the MetaCyc database because this is used in gapseq by default and was employed for all results presented in the manuscript.

- To point 1) We agree that string comparison between databases could be vague. Whenever possible, we prioritized the mapping of sequences and reactions via EC numbers because most reactions have associated EC numbers (95% for enzymatic reactions in MetaCyc). In addition, many reactions have associated Uniprot IDs already in the database (90% for reactions in MetaCyc), which circumvents the need for reaction name string matching. If string matchings could not be avoided, then the enzyme name is used to search for potential sequences in Uniprot.
- To point 2) As mentioned above, for almost all reactions a EC number can be directly found (95% for enzymatic reactions in MetaCyc). In addition, more than 70% of these reactions with EC numbers have a full and specific EC number. For the remaining reactions with unspecific EC numbers, in more than 85% a curated Uniprot association is available. For the remaining unsolved cases, matching via enzyme names is used.

Of in total >7000 enzymatic reactions with associated EC numbers in the MetaCyc database, only 1.8% reactions have more than one EC numbers associated. In these cases, gapseq checks whether for all associated EC numbers sequence evidence can be found. To the final question whether these points could lead to the incorporation of false positives, we agree that this is possible but the effects are limited, as pointed out, to narrow cases. We updated the manuscript in subsection 4.2 accordingly.

[Comment - 11]

Lines 464-472: From the total of evaluated genomes how many times per genome a pathway is predicted to be completed because of criteriums 1 (lines 466-469) or 2 (469-472) (in average)?

[Answer]

The two criteria, in which the pathway completeness threshold is lowered, account on average for around 8% of the predicted pathways per genome. In more detail, criterion 1 was fulfilled on average for 13.52 pathways ($SD = 5.03$) and criterion 2 was met for 3.28 pathways ($SD = 3.12$) per genome. In relation to the total number of predicted pathways in each genome, criterion 1 was fulfilled on average for 6.8% of the considered pathways ($SD = 2.2\%$) and criterion 2 for 1.2% ($SD = 1.1\%$).

[Comment - 12]

Lines 479-480: what happens with FASTA files that do not have that information? Some tools such as ModelSEED are able to predict protein complexes and do not need headers. They only need protein sequences. I think that the approach presented by the authors limits the application of the pipeline because there are different genome annotation pipelines and the headers in the output files of these pipelines are not homogenous among them. Therefore, you should specify the compatibility with annotation pipelines. I guess that you used the structure used in the genome annotations from NCBI.

[Answer]

It appears that in the original manuscript version our explanation of the implemented protein complex detection might have been capable of being misunderstood. In the gapseq approach, protein complexes are predicted solely on the organism's genome sequence, without the need for prior genome annotation. We extract the subunit information from the UniProt fasta headers of the reference protein sequences. These reference sequences of enzyme subunits are used for homology searches in the query genome sequence. We rephrased the subsection (4.4) "Protein complex prediction" in the revised manuscript to improve clarity.

[Comment - 13]

Lines: 484-486: "could not be matched" instead of "could not matched". In general, the paper is well written. There are a few other mistakes. Please check the manuscript thoroughly.

[Answer]

This and several other mistakes within the manuscript were corrected in the revised version.

[Comment - 14]

Lines 484-486: different verb tenses are used. "Were" should be used instead of "are" in line 485

[Answer]

Tenses were corrected.

[Comment - 15]

Lines 493-521: In many cases, TC Database has general transporters. If you reduce the TC database considering only metabolites that match those known to be produced or consumed by microorganisms, you will remove these general transporters which may also be useful. How many did you remove?

[Answer]

We do not remove transporters per se. All substances mentioned in the file `dat/sub2pwy.csv` (so far 379) were used to find corresponding transporters in the TCDB database. We also search by alternative names to recognize for example acetate and acetic acid. From the 19,129 entries in the TCDB database, `gapseq` accounts for 2,859 in a default run. However, the top five most occurring transporters in the TCDB database are: 'Predicted protein', 'Putative membrane protein', 'hypothetical protein', 'Putative uncharacterized protein', 'Uncharacterized protein'. These transporters occur several thousand times and cannot be related to transported substances. Alternatively, we provide a user-defined transporter sequences file, to add further transporter sequences (`dat/seq/transporter.fasta`).

[Comment - 16]

479-481: I think this should be evaluated. The BiGG database is a collection of models that has been manually curated. Each model has reactions with gene-protein-reaction (GPR) associations. In the BiGG database some reactions are catalyzed by protein complexes, which is represented as a GPR with the logical operator & (and). For example, the GPR for reaction GLYCTO2 (<http://bigg.ucsd.edu/models/iML1515/reactions/GLYCTO2>) is "b2979 and b4468 and b4467". What is the degree of agreement between GPRs of BiGG and your database?

[Answer]

In the case of the curated BiGG-Model for *E. coli* (iML1515), 59 reactions are associated with protein complexes and shared with all automated reconstructions using `carveme`, `modelseed`, and `gapseq`. Comparing the curated GPRs with those from the automated reconstructions showed rather unsatisfactory results: Only 6% for `carveme`, 10% for `ModelSEED`, and 19% for `gapseq` were in agreement with the GPR associations of the curated network. This indicates that the accurate GPR prediction for reactions associated to protein complexes displays a weakness in all three reconstruction tools and we aim to improve GPR-predictions within `gapseq` in upcoming versions. As for the example of GLYCTO2 (Glycolate oxidase), the GPR expressions from the `ModelSEED` and `gapseq` reconstructions were identical to those in BiGG.

Since we think this is important information to further improve automated reconstruction pipelines (including ours), we included the results from GPR comparisons in the revised manuscript (lines 236-249 (results), 624-627 (discussion), 1059-1062 (methods)).

[Comment - 17]

Line 500: determining the substrate type using the header is OK. However, as mentioned for lines 479-480, this limits the application of the pipeline.

[Answer]

We agree that searching for the transporter type using the fasta header has limits. Nonetheless, TC numbers are reasonable abstractions which are easy to detect. In general, using TC number fits well to the `gapseq` approach of identifying sequences and reactions by

EC numbers. Apart from that, the MetaCyc database also contains transport reactions with associated reference proteins which are considered in addition to the fasta header approach for TCDB entries.

[Comment - 18]

Lines 528-538: A big effort has been made by the ModelSEED team during the last years to improve and curate their database [reference 2]. We see that the ModelSEED database was retrieved more than 2 years ago (line 525). The inconsistencies mentioned by the authors and many others were corrected in the new ModelSEED database. Therefore, this is a duplication of efforts. Can the new ModelSEED database be easily incorporated into the pipeline?

[Answer]

We are pleased to see that the ModelSEED database is continuously improved as this will also further enhance the predictive potential of gapseq reconstructions. In fact, since the initial submission of our manuscript we have further expanded the gapseq reactions and metabolite database and in part also by migrating recent changes from the ModelSEED biochemistry database to gapseq. We will continue this process together with future developments of the ModelSEED database and our software.

Yet, new versions of the ModelSEED database cannot be incorporated in their entirety at once. This is because the gapseq biochemistry database is used to construct a universal network model that plays a central role in the gapfilling algorithm. In order to ensure integrity of the universal model, we test before each update if the model is still free of thermodynamically infeasible- and ATP-generating reaction cycles (subsection 4.6). In addition we omit certain reactions from ModelSEED, i.e. we generally omit aggregate reactions (e.g. <https://modelseed.org/biochem/reactions/rxn13666>), unspecific/generic reactions (<https://modelseed.org/biochem/reactions/rxn11508>), and model-specific reactions (<https://modelseed.org/biochem/reactions/rxn13254>).

In brief, corrections from the ModelSEED database are progressively transferred to gapseq, but before each update we also make sure that the changes do not entail any new inconsistencies in the gapseq database and the universal model. In general, we emphasise that the underlying biochemistry database of gapseq should be considered as a derivative of the comprehensive and already high-quality ModelSEED database. The gapseq biochemistry database is thereby maintained and continuously curated by the gapseq developer team. We have elaborated on the relationship between ModelSEED's and gapseq's database in the revised manuscript (lines 797-802).

[Comment - 19]

Lines 555-574: The approach of obtaining EC numbers, TC numbers and annotations from the genome limits the application of the pipeline as mentioned before.

The processes of pathway prediction and gap-filling seems redundant in the sense that one could avoid the process of pathway prediction and just run the gap-filling procedure after the homology search in order to create a network able to generate biomass (like in Carveme). One could just add to the draft model reactions that have a homology match and then run the gap-filling algorithm. The pathway prediction process is not perfect and therefore some reactions that shouldn't be added will be added to the draft model. Why not skip this process at all and just run a gap-filling algorithm where the minimum amount of reactions needed to make biomass are added to the network? One of the main problems with ModelSEED (at

least the old version) and CarveMe is that they return networks with too many reactions which shouldn't be in the reconstruction. It seems that the same problem is in the author's pipeline and therefore, their method is not solving the main problem of current reconstruction tools. I would appreciate your comments. If necessary, try to explain this somewhere in the manuscript.

[Answer]

As already mentioned above, the original description of enzyme and transporter prediction in the manuscript was maybe not explained unambiguous. We do not obtain EC or TC numbers nor reaction names from the genome annotation. On the contrary, gapseq does not need a genome annotation at all and rather identifies genes whose translated sequences are homologous to reference protein sequences of known function. EC and TC numbers are derived from the pathway or subsystem databases and are used to link metabolic reactants and sequence data from UniProt. We clarified the enzyme and transporter prediction in the revised manuscript (subsections 4.4 and 4.5).

We agree with the reviewer that it would be an elegant way to combine pathway prediction and gap filling. Nonetheless, our main argument against this is that we attempt to minimise the influence of the gap filling medium, which represents one of the most influential assumptions of the overall approach. Often (especially in metagenomic data), the chemical composition of the growth environment or isolation medium is unknown or rather vague. To focus on a gap filling step alone would therefore risk introducing a strong bias due to the potential uncertainty of the gap filling medium. Therefore, our approach combines a medium-independent pathway prediction with a medium-dependent gap filling.

We are of the same opinion that it would be desirable to have metabolic reconstructions with a minimum number of reactions that are added to the network without evidence from sequence homology to reference protein sequences. In this context we would like to emphasize that the gap-filling steps 2 to 4 only add reactions which

- (i) have sequence evidence, i.e. whose homology to reference proteins are significant, although with a bitscore below the threshold to be included directly in the draft network and
- (ii) which enable metabolic functions that are potentially relevant in nutritional environments different to the user-provided gap-filling medium or in the organism's engagement in metabolic interactions with other organisms/models in downstream microbial community simulations.

In addition, also the first gap-filling step (enabling biomass formation) minimises the number of required gapfill reactions without sequence support by assigning those reactions a maximum weight/penalty in the LP-based optimisation.

In summary, we consider the incorrect representation of an organism's physiology (e.g. fermentation by-products; interactions with other organisms) as the main problem of current reconstruction tools (as now pointed more clearly in the introduction, lines 45-90). The tools ModelSEED and CarveMe displayed shortcomings in the accuracy of physiological predictions as shown for instance in Figure 3 and 4. We argue that these problems could be attributed to the strong impact of the gap filling medium and which was our motivation to implement a gap filling approach that reduces the gap-fill medium effects.

We recognise that the ideas behind our approach to gap filling needed further clarification. Therefore, we have added a new paragraph to subsection (4.9.) in the methods section.

[Comment - 20]

575- 610: the gap-filling algorithm is quite clever.

Thank you.

[Comment - 21]

618-619: Here, it says that the gap-filling procedure is performed in 4 steps. I don't understand why you need four steps. the goal of the gap-filling procedure is just one: to achieve biomass production. Steps 2 to 4 are not necessary.

[Answer]

We respectfully disagree with the reviewer on this point. The different steps of the gap filling algorithm follow quite different approaches. On one hand, we have a 'classical' gap-filling in step 1 which enables the capacity to produce biomass given a growth medium. This step uses all available reactions from gapseq's reaction database. On the other hand, we have growth medium-independent steps 2-4, which are designed to enhance the predictive power of the metabolic model without relying too much on growth conditions. For this reason, the gap filling here is only done by using reactions, which have sequence evidence but are not yet part of the metabolic model. For example, step 2 checks whether individual biomass components could be formed given a minimal medium that does not already provide the specific biomass components. Here, the objective for optimisation is set to maximise the production of the individual biomass component and only those reactions are gap-filled that were found by '*gapseq find*' based on sequence homology. The rationale behind this is that a rich growth medium, which contains for example amino acids, would not necessarily lead to a gap-filled model (step 1) which can synthesize these amino acids even if the organism possesses this anabolic capacity. Therefore, the biosynthesis of each biomass component is checked in step 2 but only by using sequence-supported reactions. The other steps 3-4 work similarly. The BIOLOG-like carbon source screening and the by-product screening are meant to enable likely metabolic phenotypes based on sequence evidence without introducing a bias caused by a specific gap-filling medium definition. The description of the gap-filling procedure was revised in the manuscript (subsection 4.9) in order to improve clarity in gapseq's approach to this crucial part of automated model reconstruction of gap-filling.

[Comment - 22]

629-631. With this approach, you could add more reactions than the ones actually needed. Again, wouldn't the networks created with this pipeline have too many reactions? On average, how big are the genome-scale metabolic reconstructions built with this pipeline? Do users have to define which alternative energy sources can be used by the microorganism? If the user does not provide data, then how the pipeline decide which carbon sources should sustain in silico growth? This should only be performed if the user provides data. Otherwise, it would lead to false-positive results. Why M9 media is used? Shouldn't you use a user-defined media? Why this is performed only for carbon sources and not for nitrogen sources?

[Answer]

We agree with the reviewer that the corresponding section 4.9 needed some clarification. Most importantly, the reactions which were added in gap-filling steps 2-4 are only those which have direct sequence support but were not added to the draft model yet. By this, our approach can enable the prediction of carbon source utilisation and production of metabolic by-products while reducing the risk to include false positive reactions. On average, gapseq models are larger than ModelSEED models but similar in size compared to CarveMe (BacDive genome set; Mean number of reactions 1210 (ModelSEED) , 1657 (CarveME), 1691 (gapseq)). In the revised manuscript we rephrased and extended section 4.9 to improve clarity in the description of the gap-filling algorithm.

[Comment - 23]

Line: 637: what do you mean with artificial? That's not clear. Please, specify.

[Answer]

The corresponding reactions are added temporarily to the model to check whether loaded reducing equivalent carriers (NADH, ubiquinol, menaquinol) could be recycled by these reactions. We updated the manuscript in section 4.9 to clarify it.

[Comment - 24]

Line 642: what do you mean with "the same list of compounds as for step (3)"? Are you referring to ubiquinol, menaquinol, or NADH? In general, this part of gap-filling is not very clear. Please, provide more details.

[Answer]

The same list of compounds refers to the list of metabolites as defined in dat/sub2pwy.csv. We added this to the sentence. Throughout this review, the whole section 4.9 was rewritten and now provides a more detailed description.

[Comment - 25]

Lines 678-680: the sentence is strangely written. Maybe you could re-phrase it.

[Answer]

We updated the sentence, thank you for this suggestion.

[Comment - 26]

678-688: Could this approach be biased? I wonder how complete are the annotations (in terms of EC numbers) for the model files created with Carveme and ModelSEED. The authors map reactions through EC numbers. What happens if a model file created with Carveme does not have many annotations in terms of EC numbers. Then, the match between the EC numbers in the genome and the metabolic model would be very poor and the conclusion obtained with the author's pipeline would be that the model generated with Carveme is very poor. However, the reality is that reactions are there, but the annotations (EC numbers) are missing. Does the pipeline of the authors does consider this type of bias? Please comment.

[Answer]

We considered only those reaction activities from the BacDive database for the enzymatic data test for which an EC matching to all reaction namespaces (BIGG and ModelSEED) of all three reconstruction tools were available. In addition, the mapping of EC numbers to reaction IDs is the same for ModelSEED and gapseq. We added additional text to section 4.11 to clarify this.

[Comment - 27]

685-688: I don't understand what kind of data is present in BacDive. In line 664, you only mentioned that you used enzymatic data, but readers would appreciate that you could specify which kind of data can be found in that database. What "active" and "not active" mean? Is this measured enzymatic activity experimentally collected for strains grown on a particular media condition?

[Answer]

We updated the results subsection for the enzymatic data tests (4.11) and elaborated on the origin and type of data we used for evaluation. Please also refer to our detailed response to comment 34 below, which also correspond to the reference data from BacDive and how we improved the manuscript in this matter accordingly.

[Comment - 28]

Line 699-701: I don't understand why you removed D-lyxose. If all the pipelines predict no utilization and that agree with experimental data, it is OK to include it.

[Answer]

We removed D-lyxose because it was largely overrepresented in the source database (500 tests). Beyond that, all experimental growth data for D-lyxose was negative and all three pipelines (CarveMe, ModelSEED, gapseq) predicted no growth. Thus, we couldn't find any discriminative power to be associated with the addition of D-lyxose to the carbon source test. Ultimately, we decided to exclude D-lyxose because barplots in Figure 1B would become non-readable because of the large part of true negatives for all pipelines.

[Comment - 29]

Lines 702-704: This seems very important to understand the performed in silico test. It is very interesting the approach that you used because it is indeed closer to the experimental procedure of BIOLOG arrays than when you maximize growth rate. Can you explicitly describe the reaction equations of the reactions that you temporarily added to the model?

[Answer]

Thank you for pointing to the importance of the in silico test that was used to validate the prediction of carbon source utilisation. The reactions, which were added temporarily, are recycling reactions of compounds, which carry reducing equivalents. In more detail, the reducing equivalents of quinol, menaquinol, and NADH were discharged (ESP1: ubiquinol → ubiquinone + 2H⁺; ESP2: menaquinol → menaquinone + 2H⁺; ESP3: NADH → NAD⁺ + H⁺). By this, the capacity of a growth compound to function as electron donor can be tested. We took this as a proxy if this compound could be a potential carbon source. We updated the manuscript and explicitly describe this in subsections 4.9 and 4.12.

[Comment - 30]

Line 722: how changes in this value affect the results?

[Answer]

We have re-run the gene-essentiality analysis with two additional cut-off values for the minimum growth: 0.05 and 0.001 hr⁻¹ in addition to the original value of 0.01 hr⁻¹. Altering this cut-off value did yield virtually no differences in the gene essentiality benchmarks with the only exception of *M. genitalium*. In this case the gapseq prediction scores for sensitivity, accuracy, and F1-Score were slightly reduced with the cutoff 0.001, but were still higher than the scores from the CarveMe reconstruction and similar to the scores of the ModelSEED reconstruction. The essentiality tests were not feasible with the cutoff = 0.05 for the CarveMe reconstruction in the case of *M. genitalium*, as the predicted growth without knockouts was already below this threshold (0.034 hr⁻¹). We have added a supplemental figure for this parameter test (Figure S1) and a description of the results to the main text (lines 232-235).

[Comment - 31]

728-750: this looks OK.

772: Is this a dynamic simulation?

how the authors modeled the microbial communities? I think a method section describing the details for the community modeling is missing.

[Answer]

We used the framework BacArena for dynamically modeling the microbial communities. In the revised version of our submission, we updated the corresponding text (lines 1117-1135) and included all source code which can be used to reproduce the corresponding simulation and figure on the github repository (<https://github.com/Waschina/gapseqEval>).

[Comment - 32]

I would suggest including an introductory paragraph in each section of methods. These paragraphs should mention the general idea of the section.

[Answer]

The manuscript was revised comprehensively. Especially in the method section many additions and clarifications were made. We added introductory parts to the methods sections to provide a clearer structure to readers.

RESULTS

[Comment - 33]

Lines 85-86: Why you removed dead-end metabolites and the corresponding reactions. Even though reactions that have unconnected metabolites do not carry flux, they could be important to complete models as knowledge-bases.

[Answer]

Dead-end metabolites are not removed from the universal model nor from specific gapseq reconstructions. We stated the numbers in the results sections solely for the purpose to

provide networks metrics that help to roughly estimate the dimensions of the universal model and the gapseq biochemistry database. We agree that the original phrasing of this sentence might have been misleading. The wording is improved in the revised manuscript (lines 134-138).

[Comment - 34]

Lines 94-95: these 10.538 enzyme activities correspond to what? Do these 10538 enzyme activities comes from different microorganisms? Please specify in the method section (and it would be also useful in the results section) for which organisms and which conditions these enzymatic activities were collected. How many EC numbers are related to those 10538 enzymatic activities?

[Answer]

The >10,000 enzyme activities used for benchmarking correspond to laboratory microbial enzyme tests, which are frequently used for the characterisation and identification of microorganisms. In those tests, microbial cell cultures or extracts from the cultures are exposed to usually high concentrations of the substrate of the focal enzyme. For instance in the catalase (EC 1.11.1.6) test, cells of a bacterial colony are transferred and smeared onto a droplet of hydrogen peroxide. If bubbles are formed, the isolate is considered *catalase positive* as this indicates the formation of water and oxygen; the products of the catalase reactions.

BacDive is, to our best knowledge, the largest curated database for results from laboratory enzyme activity tests. In our study, we used BacDive data from 3017 organisms including 30 different enzymes. The culture conditions can vary between the microorganisms tested and the specific enzyme test. BacDive does not provide information on the exact culture conditions under which the enzyme activity was tested, but provides references to the original publications. Yet, in the course of this study, it was not feasible to extract the individual culture conditions from the thousands of publications referenced by BacDive. However, the experimental enzyme activity test conditions are generally designed to invoke the expression of the specific enzyme, if the organism harbours the respective gene(s) – especially also because these enzyme tests are often applied in clinical practice to distinguish pathogenic from commensal bacteria. Hence, for our benchmark using the BacDive data on enzyme activities, we argue that in the case of a positive enzyme activity test, the corresponding reactions should be part of the reconstructed network model. In contrast, a negative enzyme activity test provides strong evidence that the organism does not possess the respective metabolic capability and the corresponding reaction should not appear in the model reconstructions.

We recognise that the description of the rationale behind this reconstruction evaluation approach required additional information in the manuscript's results and methods sections. We revised the manuscript accordingly (subsections 4.11 and 2.2) and provided the data files and script to perform the enzyme activity benchmark on a public github repository (<https://github.com/Waschina/gapseqEval>). In addition, the new supplemental Table S7 lists all microorganisms for which BacDive data was retrieved and includes the corresponding RefSeq assembly IDs, that were used for the automated model reconstructions.

[Comment - 35]

Line 100: How do you define false negative? I guess that EC number present in the genome annotation but not present in the metabolic model?. You should explicitly define this in the method section.

[Answer]

Here, false negative means that the enzymatic activity was measured experimentally but the corresponding reaction was not found in the metabolic network. We updated the method section 4.11 to clarify this (lines 994-998).

[Comment - 36]

Line 102: this is very strange. 26% of the comparisons means $10,538 \times 26/100 = 2739$ comparisons. That means that almost all the microorganisms had the enzymatic activity 1.11.1.6. Is that correct? I wonder about the distribution of EC numbers counts (sorted frequency of EC numbers). It seems that it could be biased for certain EC numbers. How this bias could affect the results?

[Answer]

We thank the reviewer for this comment. It is true that the EC number count distribution is not uniform. As mentioned in the manuscript, cytochrome oxidases and catalases account for roughly 50% of the tests. We performed an additional check by re-sampling of the enzymatic data for 15 EC numbers which have data in BacDive for at least 100 organisms. In order to correct for potential bias, we randomly chose 100 tests for each EC number and recalculated our evaluation 500 times. We found the numbers of true positives, true negatives, false positives, and false negatives to be stable with respect to the comparison between the reconstruction tools (new Figure S4): Gapseq still showed the best sensitivity, higher than CarveMe and ModelSEED, and the specificity of CarveMe is again slightly higher than that of gapseq and ModelSeed. In comparison to the original approach, which used all enzymatic data for validation, gapseq's sensitivity is a bit lower, whereas gapseq's specificity is higher. We added text to the manuscript in section 2.2 (lines 998-1003) and added a supplementary figure (S4) which shows the comparison of original data to uniform re-sampling analysis. We kept the original figure in the manuscript and referred to the new figure in section 2.2 by arguing that the overall results (highest accuracy with gapseq models) remained stable after correcting for the potential bias due to non-uniform distribution of EC-numbers in the reference data from BacDive.

[Comment - 37]

Lines 113-120: you describe in the methods section that you used data from ProTraits. However, you didn't describe what type of data is in this database. Do the data from ProTraits come from BIOLOG arrays? Was this data collected assessing growth on flasks by measuring OD? Was this data collected using both techniques? You implemented an in silico test which is analogous to the BIOLOG test. You can only compare results from this in silico test with data that was collected with BIOLOG arrays. One can infer from the explanation that you gave for formate overestimation in lines 113-120 that the experimental data for formate was a measure of formate being incorporated into the biomass components (for example one could test this in a flask with formate as sole carbon source and by measuring OD). If that's the case, you should perform an in silico test where you actually maximize growth rate and not on the basis of electron transfer.

[Answer]

The comparison and validation of carbon source predictions is usually hindered by the incomplete knowledge of the corresponding growth conditions. Thus, the ability of automatic reconstructions of genome-scale metabolic models to predict carbon source utilisation has so far not been evaluated on a larger scale. We used the ProTraits database, which contains diverse phenotypic data that is inferred from the scientific literature and comparative genomics. Data in ProTraits include also carbon source utilisation phenotypes. However, it does not state whether the phenotype information is based on BIOLOG-like tests or actual bacterial growth experiments.

The underlying idea why we used an *in silico* BIOLOG-like test is to enable the large-scale evaluation of carbon source utilisation by using a simulation approach that does not require to define all nutritional factors that are required for cell growth (e.g. minerals, vitamins, amino acids to overcome auxotrophies), but by focusing on the main source of carbon and energy for cellular metabolism. Thus, the objective function in the BIOLOG-like approach (electron transfer from the focal substrate to reducing equivalents) requires less nutrients from the environment than the flux through the biomass reactions in growth simulations. In addition, predicting growth using a rich medium that enables a flux through the biomass reaction entails the shortcoming that the models often use other substances from the growth medium than the actual supplemented compound, whose ability to serve as a carbon source is anticipated to be tested.

The BIOLOG-like approach therefore enabled to targetly and uniformly test individual carbon source utilisation capacities across the wide range of organisms included in ProTraits. In this context, the example of formate is especially relevant because it shows the limit of the approach. For most common carbon sources, it is true that they can also act as electron donors. Nonetheless, there are exceptions like formate, which has been described to frequently serve as electron donor for several bacteria, but only in a few cases as actual source of carbon for cell growth. This is because the formation of carbon-carbon bonds from C1 compounds is energetically challenging. Interestingly, gapseq shares this limitation with BIOLOG, which is nonetheless commonly applied as indirect test or proxy for carbon source utilisation. However, formate represents only 1 out of 48 total unique carbon sources within the ProTraits evaluation data set.

We revised the respective parts in the manuscript by elaborating on the description of the actual validation data from ProTraits, our approach to test carbon source utilisation, and by clearly stating the advantages and limitations of this approach (subsections 2.3 and 4.12).

[Comment - 38]

Results from sections 2.4 and 2.5 are very impressive. It is quite shocking how bad carveme and modelseed performs in terms of by-product prediction.

Lines 174-176. You should briefly describe the community modeling framework BacArena. Some basic questions that you should answer are

- 1) Is this a dynamic simulation or is a steady-state simulation? It seems that it is dynamic because you mention time step in line 175 but you should make this explicit.
- 2) What was the objective function of the microbial community? Or each model was optimized separately in a common environment?
- 3) What are the inputs and outputs of this framework?

[Answer]

Thank you for your interest in the community framework that we used for the anaerobic food web simulation. BacArena is an agent-based framework that employs flux balance analysis to simulate microbial communities in time and space. We performed a dynamic simulation and since all organisms were represented as individuals, the corresponding objective functions were kept as biomass optimisation so that each model was optimised separately in a shared growth environment. As input, BacArena requires metabolic models and information about substance availability. The output is a simulation file, which consists of properties of individuals (position, biomass, uptake rates, reaction activities) and substances (spatial distribution of concentrations) for each time point. For a more detailed description we would like to link to the original BacArena publication (<https://doi.org/10.1371/journal.pcbi.1005544>). The manuscript was updated to improve the manuscript by providing more details on the BacArena simulations as suggested (lines 1117-1135 and 288-296).

[Comment - 39]

Looking at methods (lines 775-777), why did you choose the third time step? It would be good that you actually show the predicted uptake and production for each time step until the final hour. Why you show uptake and production rates and not changes in metabolite concentrations? This would make more sense as it is a dynamic simulation.

[Answer]

We choose time step three because it corresponds to the exponential growth phase of the community simulation before the inflection point. Nonetheless, we found the overall growth rate to be lower for ModelSEED in comparison to gapseq and CarveMe models. This is why we updated our results and selected time step five for ModelSEED. In time step five the ModelSEED community reached a similar total biomass as the gapseq and CarveMe communities in time step three. This modification did not alter the overall results. To better illustrate the BacArena simulation, we have integrated community growth curves in Figure 4. Moreover, we included the new Figure S5, which visualised the community simulation results (production and consumption rates) for all time steps as suggested by the reviewer. We chose heatmaps to visualise the community metabolism and not the changes in metabolite concentrations, because the organism-X-metabolite heatmaps allow to visualise the contribution of each individual organism/model to the shared chemical growth environment.

[references highlighted by reviewer #1]

References

- 1) Overbeek, R., Begley, T., Butler, R. M., Choudhuri, J. V., Chuang, H. Y., Cohoon, M., de Crécy-Lagard, V., Diaz, N., Disz, T., Edwards, R., Fonstein, M., Frank, E. D., Gerdes, S., Glass, E. M., Goesmann, A., Hanson, A., Iwata-Reuyl, D., Jensen, R., Jamshidi, N., Krause, L., ... Vonstein, V. (2005). The subsystems approach to genome annotation and its use in the project to annotate 1000 genomes. *Nucleic acids research*, 33(17), 5691-5702. <https://doi.org/10.1093/nar/gki866>
- 2) Samuel M. D. Seaver, Filipe Liu, Qizhi Zhang, James Jeffryes, José P. Faria, Janaka N. Edirisinghe, Michael Mundy, Nicholas Chia, Elad Noor, Moritz E. Beber, Aaron A. Best, Matthew DeJongh, Jeffrey A. Kimbrel, Patrik D'haeseleer, Erik Pearson, Shane Canon, Elisha M. Wood-Charlson, Robert W. Cottingham, Adam P. Arkin, Christopher S. Henry

bioRxiv 2020.03.31.018663; doi: <https://doi.org/10.1101/2020.03.31.018663>

Reviewer #2

[Comment - 40]

The manuscript entitled "gapseq: Informed prediction of bacterial metabolic pathways and reconstruction of accurate metabolic models" by Johannes Zimmermann, Christoph Kaleta and Silvio Waschina, is a computational paper focused on the development of a software that assists in the automated reconstruction of genome-scale metabolic models, integrating a vast amount of information collected through databases.

This is a solid work, useful for the community, that presents a new approach to overcome the major pitfalls of the current genome-scale metabolic reconstruction approaches. Many details are reported, to inform the reader about how the pipeline of the work has been envisioned, which data have been retrieved, and how these were used. The manuscript is very well written, with concepts being clearly presented.

Through this elegant effort, metabolic models that are reconstructed using gapseq may be used to generate models where multiple species co-exist, such the environment observed in the microbiome.

[Answer]

We thank the reviewer for the careful evaluation and suggestions for improvements. Please find below how we revised our manuscript in response to the individual comments. Indicated line numbers refer to the position in the revised manuscript with tracked changes.

[Comment - 41]

Below I highlight a couple of major point for improvement as well as some minor comments, which should be addressed before publication:

Major comments:

- Page 3, line 78. The author state that "In addition, the protein sequence database in gapseq can be updated to include new sequences from Uniprot and TCDB." Is the update automatic or manual? To make the software useful for the community, the update should be automatic, so that the most recent updates may be available when the user launches a job through the software.

[Answer]

We thank the reviewer for the idea that an automatic update function should be available. We integrated this into *gapseq* (git commit: bb65733) and rephrased the respective sentence in section 2.1 of the manuscript.

[Comment - 42]

- Page 7, line 176. The authors state that "On the community level, simulations using gapseq models captured all important substances, which are known to be produced in the context of the food web (Figure 4)". While gapseq largely outperforms regarding the production of important substances, the consumption appears to be not recapitulated by gapseq. However, the authors do not highlight this and, at Page 7, line 213 they state that "The overall consumption pattern and individual microbial contributions were found to be in agreement with literature data.". This statement is not supported by the outcome of Figure 4,

and the authors should develop further on the outcome presented in Figure 4 regarding the consumption across CarveMe, gapseq and ModelSEED, and identify the reasons of the decrease of gapseq in capturing the consumption.

[Answer]

We agree with the reviewer that not all metabolite consumption features that are expected (based on literature information from experimental studies) are recapitulated in our community metabolism simulation using gapseq models. While *gapseq* models performed well with regard to the consumption of lactate and succinate in the community simulation, the cross-feeding (i.e. the uptake) of acetate and hydrogen produced by other community members were not predicted for other utilisers than *M. barkerii*. In the revised manuscript, we elaborated on the substance consumptions that we correctly predicted in the community simulation and on those which were expected but not predicted (lines 328-352). The causes of missing predictions are now further investigated. In brief, we observed that strains, which are known to use low energy-yielding substrates (such as acetate) that are produced by other community members, utilised saccharides (i.e. glucose and fructose) as main resource for growth instead.

Also the concluding statement in this subsection was rephrased to be more balanced between correct and incorrect/missing predictions shown in Figure 4 (lines 362-368).

[Comment - 43]

Minor comments:

- Page 2, line 16. The authors state that "Metabolic models not only have demonstrated their ability to predict phenotypes on the level of cellular growth and gene knockouts, but also provide potential molecular mechanisms in form of gene and reaction activities, which can be validated experimentally [5]". The reference is outdated; following the reasoning of the authors, it would be appropriate to indicate more recent modelling efforts which have led to an experimental validation of some among those molecular mechanisms.

[Answer]

The references for this statement were updated as suggested in the revised manuscript.

[Comment - 44]

- Page 3, line 66. The authors state that "Topology as well as sequence homology to reference proteins inform the filling of network gaps, and the screening for potential carbon sources and metabolic products is done in a way that reduces the impact of growth medium definitions." It would be better to indicate, shortly, how this is done, rather than state "...is done in a way that...".

[Answer]

The respective part was rephrased as suggested (lines 100-111).

[Comment - 45]

- Page 9, line 258. The authors state that "For each validation approach, predictions were compared to experimental data obtained from databases and literature to calculate prediction performance scores.". It would be useful to mention which type of data from the

literature and which databases have been used for the validation, so that the reader will be immediately aware - here in this paragraph - of where to retrieve the relevant information.

[Answer]

The paragraph of subsection (2.9) was extended as suggested by the reviewer (lines 417-432).

[Comment - 46]

- For some reason, the references do not appear in the text in a progressive numbering. This should be fixed.

[Answer]

We apologise for this mistake and corrected the reference numbering in the revised manuscript.

[Comment - 47]

- Some of the articles listed in the references include information that do not pertain to the article(s), such for example the manuscripts cited that have appeared on Current Opinion in Systems Biology. The correctness of the information should be double-checked.

[Answer]

Corrected in revised manuscript version.

[Comment - 48]

- Figure S2 is indicated in the main text, but Figure S1 is not.

[Answer]

Figure S2 (previously Fig. S1) is now referenced in subsection 2.6.

[Comment - 49]

- Typos:

- Page 5, line 126. Please, correct "essentially" with "essentiality" .
- Page 5, line 135. Please, remove the comma after "...to the fact".
- Page 5, line 145. "Figure 2" should be "Figure 3".
- Page 13, line 369. Please, remove the comma after "Although".

[Answer]

Typos are corrected in the revised manuscript.

Second round of review

Reviewer 1

Thanks for taking care of all the comments that I did. The answers provided useful details and helped to understand the procedures. Only minor concerns still remain.

Regarding comment 21, you say that "For this reason, the gap filling here is only done by using reactions, which have sequence evidence but are not yet part of the metabolic model".

".. and only those reactions are gap-filled that were found by 'gapseq find' based on sequence homology"

This seems strange to me as reactions with sequence evidence should be already part of the draft model and they shouldn't be used to gap-fill the model as there is already enough evidence to include them in the draft model before the gap-filling process. Does sequence evidence mean homology evidence, right? It seems that not all the reactions with homology evidence are included in the draft model. During the process of draft generation, you include reactions from the pathway prediction and the transporter prediction. In section 4.7 it says "A reaction is added to the draft model if the corresponding enzyme/transporter was directly found or if the pathway was predicted to be present". Thus, I wonder how many reactions (in average) do have homology evidence and are not included in the draft model (before the gap-filling)? I would be very helpful for readers and potential users to know how reconstruction sizes change after each procedure. For example. You start with a draft network of 1000 reactions after pathway and transporter prediction. Then, after the first step of gap-filling, you add 100 reactions. Then after the second step of the gap-filling you add 50 more, etc. etc. Do gapseq generate during the reconstruction process an automatic report so the user can keep track of the changes in the model during the different stages?

The revised version is clearer than the previous one; readers will appreciate it. I think the manuscript is almost ready to be published as it provides a novel method to the community, sufficiently explains how it works, why users should choose it, and its limitations.

I have still one general concern regarding the method: it doesn't tackle down the false-positive results (probably due to the inclusion of too many reactions), which are usually high in tools such as CarveMe or ModelSEED (compared with manually curated models). However, the manuscript transparently shows this in figure 1, and it is also mentioned in the discussion and in the answers. Therefore, I think the method is sufficient as it is.

Reviewer 2

The authors have addressed all critical points that I have raised in the first round of revision.

In particular:

- The automatic update function is now available for gapseq;
- A more appropriate balance of the correct versus incorrect/missing predictions presented in Figures 4 and S2 is detailed in the text of Section 2.6;
- Paragraph 2.9 has been extended, to include the literature data and databases that were used for validation.

I am happy to recommend this elegant and solid work for publication. The tool developed by the authors will be useful for the scientific community working on genome-scale metabolic reconstructions.

Response to reviewer

[Reviewer #1 comment]

Dear authors,

Thanks for taking care of all the comments that I did. The answers provided useful details and helped to understand the procedures. Only minors concerns still remain.

Regarding comment 21, you say that "For this reason, the gap filling here is only done by using reactions, which have sequence evidence but are not yet part of the metabolic model".

".. and only those reactions are gap-filled that were found by 'gapseq find' based on sequence homology"

This seems strange to me as reactions with sequence evidence should be already part of the draft model and they shouldn't be used to gap-fill the model as there is already enough evidence to include them in the draft model before the gap-filling process. Does sequence evidence means homology evidence, right? It seems that not all the reactions with homology evidence are included in the draft model. During the process of draft generation, you include reactions from the pathway prediction and the transporter prediction. In section 4.7 it says "A reaction is added to the draft model if the corresponding enzyme/transporter was directly found or if the pathway was predicted to be present". Thus, I wonder how many reactions (in average) do have homology evidence and are not included in the draft model (before the gap-filling)? I would be very helpful for readers and potential users to know how reconstruction sizes change after each procedure. For example. You start with a draft network of 1000 reactions after pathway and transporter prediction. Then, after the first step of gap-filling, you add 100 reactoins. Then after the second step of the gap-filling you add 50 more, etc. etc. Do gapseq generate during the reconstruction process an automatic report so the user can keep track of the changes in the model during the different stages?

[Authors reply]

We thank the reviewer for this comment. In our previous response to comment 21, we referred to reactions for which BLAST hits from reference proteins to the query genome were found, but for which the bitscore of the alignment did not reach the threshold to include the reaction directly in the draft network. By default, gapseq uses a bitscore of 200 for this cutoff value (parameter 'b'), but it can also be adjusted by the user. Reactions having lower alignment bitscores are not directly included in the draft network to prevent large networks

that could include too many false positive reactions. However, reactions with an alignment bitscore lower than 'b' are considered for the gap-filling algorithm as candidate reactions. Reactions with the highest bitscores are preferred over reactions with lower alignment bitscores in order to quantitatively reflect different levels of sequence evidence. We updated the manuscript (lines 777-780) to further clarify this procedure.

The number of reactions that are added to the network model at the different steps of the reconstruction process can be tracked by the user in two ways. First, gapseq prints messages to the standard command line output, which states how many reactions are added at the individual gap-filling steps. Second, detailed information for each reaction, which indicates why a reaction has been included in the network model, is directly stored in the final model file. These model files contain a table, where each row represents a model reaction. The column 'gs.origin' states why each reaction was added to the model and at which step. The documentation of gapseq includes a tutorial of how to access this information (<https://gapseq.readthedocs.io/en/latest/tutorials/traceability.html>) and we have now included additional information on reaction traceability during reconstruction in the manuscript at lines 895-905. On the basis of models reconstructed for the manuscript, gapseq has added on average 1297 reactions directly to the draft network, and 140 reactions during gap-filling (step 1: 115 reactions; step 2: 3; step 3: 15; step 4: 7); however, the numbers can highly vary depending on the input genome sequence and gap-filling medium.